# Tumor protein D54 defines a new class of intracellular transport vesicles

Gabrielle Larocque, Penelope J. La-Borde, Nicholas I. Clarke, Nicholas J. Carter, and Stephen J. Royle

**Transport of proteins and lipids from one membrane compartment to another is via intracellular vesicles. We investigated the function of tumor protein D54 (TPD54/TPD52L2) and found that TPD54 was involved in multiple membrane trafficking pathways: anterograde traffic, recycling, and Golgi integrity. To understand how TPD54 controls these diverse functions, we used an inducible method to reroute TPD54 to mitochondria. Surprisingly, this manipulation resulted in the capture of many small vesicles (30 nm diameter) at the mitochondrial surface. Super-resolution imaging confirmed the presence of similarly sized TPD54-positive structures under normal conditions. It appears that TPD54 defines a new class of transport vesicle, which we term intracellular nanovesicles (INVs). INVs meet three criteria for functionality. They contain specific cargo, they have certain R-SNAREs for fusion, and they are endowed with a variety of Rab GTPases (16 out of 43 tested). The molecular heterogeneity of INVs and the diverse functions of TPD54 suggest that INVs have various membrane origins and a number of destinations. We propose that INVs are a generic class of transport vesicle that transfer cargo between these varied locations.**

## Introduction

Eukaryotic cells are by definition compartmentalized: they contain organelles and membrane-bound domains that have distinct identities. Vesicle transport between these locations is tightly regulated to maintain these identities, yet allows exchange of specific materials. There are several types of vesicular carrier described so far that are classified according to morphology or location. Well-characterized examples include clathrin-coated vesicles (50–100 nm diameter) formed at the plasma membrane (PM) or TGN, COPII-coated vesicles (60–70 nm) originating at the ER, and intra-Golgi transport vesicles (70–90 nm; Vigers et al., 1986; Balch et al., 1994; Orci et al., 2000). Whether cell biologists have a complete inventory of vesicular carriers is an interesting open question.

In humans, there are four tumor protein D52-like proteins (TPD52-like proteins; TPD52, TPD53/TPD52L1, TPD54/TPD52L2, and TPD55/TPD52L3), some of which have been associated with membrane trafficking, but the cell biological roles of the family are not well characterized. TPD52-like proteins are short (140–224 residues), have 50% identity, and each contain a coiled-coil domain through which they can homodimerize or heterodimerize (Byrne et al., 1998). All are ubiquitously expressed with the exception of TPD55, which is restricted to testis (Cao et al., 2006). TPD52 was the first of the family to be identified due to its overexpression in cancer, and it is still the best studied. However, all members have been found to be overexpressed in a series of cancers (Cao et al., 2006; Byrne et al., 1995, 1998; Nourse et al., 1998). Overexpression of TPD52 correlates with poor prognosis in breast cancer patients, and in cell models, TPD52 overexpression promotes proliferation and invasion (Byrne et al., 2010, 1996; Li et al., 2017; Dasari et al., 2017).

Rather disparate membrane trafficking functions have been reported for TPD52 and TPD53. First, TPD52 is involved in secretion in pancreatic acinar cells (Thomas et al., 2004, 2010; Messenger et al., 2013) and potentially at synapses (Biesemann et al., 2014). Second, membrane trafficking proteins bind to TPD52, such as the endocytic protein Rab5c (Shahheydari et al., 2014), and the transcytotic protein MAL2 (Wilson et al., 2001). Third, TPD52 has a role in lipid droplet biogenesis at the Golgi (Kamili et al., 2015; Chen et al., 2019). Finally, a role in membrane fusion was proposed for TPD53 (Proux-Gillardeaux et al., 2003). By contrast, the potential functions of TPD54 remain unexplored.

What is striking about TPD54 is its sheer abundance in cells. Previous quantitative proteomic analyses revealed that TPD54 is one of the most abundant proteins in HeLa cells, ranked 180th out of 8,804 (Hein et al., 2015; Kulak et al., 2014). There are an estimated $3.3 \times 10^6$ copies of TPD54 per HeLa cell (2.7 µM), whereas abundant membrane traffic proteins such as clathrin light chain A or β2 subunit of AP2 total $2.2 \times 10^6$ or $1.0 \times 10^5$ copies (1.8 µM or 0.4 µM), respectively (Hein et al., 2015).

Centre for Mechanochemical Cell Biology, Warwick Medical School, University of Warwick, Coventry, UK.

Correspondence to Stephen J. Royle: s.j.royle@warwick.ac.uk

A preprint of this paper was posted in *bioRxiv* on August 12, 2019.

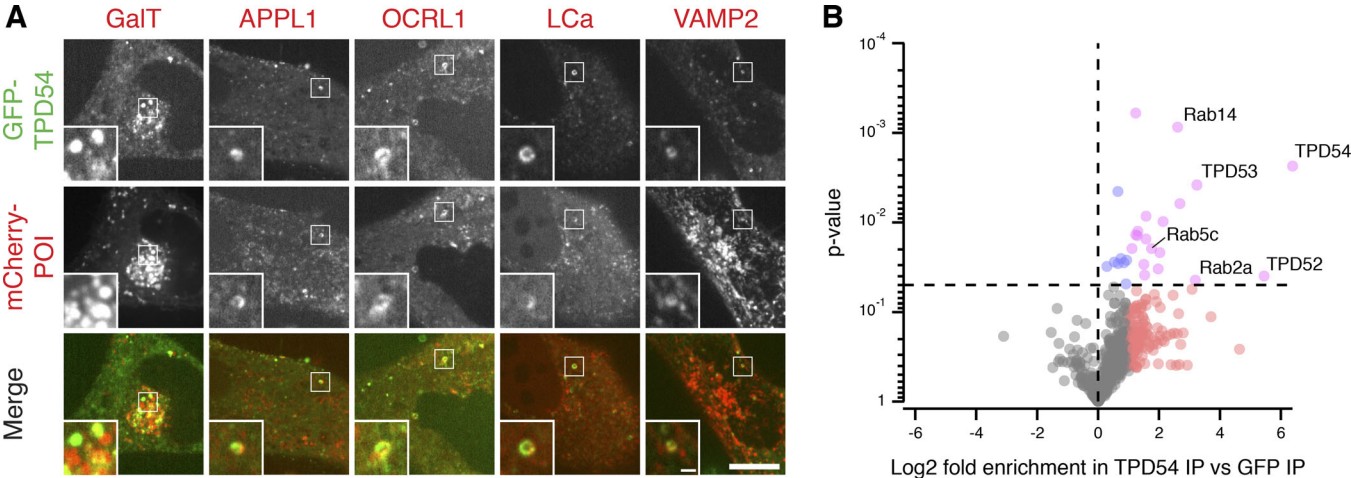

Figure 1. **TPD54 is a membrane trafficking protein. (A)** Representative confocal micrographs showing transiently expressed mCherry-tagged membrane trafficking proteins of interest (POI) and endogenously tagged GFP-TPD54. Inset, 3× zoom. Scale bars, 10 µm, 1 µm (inset). **(B)** Volcano plot of a comparative mass spectrometry analysis of GFP-TPD54 vs. GFP co-immunoprecipitation. Proteins enriched more than twofold in GFP-TPD54 samples compared with GFP are shown in red or pink; those P < 0.05 are shown in blue or pink. $n_{exp}$ = 4. Note, glycogen debranching enzyme (3.7-fold increase, P = 1.09 × 10$^{-8}$) is not shown. Proteomic data and volcano plot calculations are available (Royle, 2019). IP, immunoprecipitation.

Despite its abundance, there are virtually no published data on the cell biology of TPD54. Due to sequence similarity and heterodimerization properties, we hypothesized that TPD54, like the other members of the family, would also be involved in membrane trafficking. We set out to investigate the cell biology of TPD54 and found that it defines a novel class of intracellular transport vesicle, which we have termed intracellular nanovesicles (INVs). These vesicles are small, functional, and molecularly diverse, suggesting that they mediate transport throughout the membrane traffic network.

## Results

### TPD54 is a membrane trafficking protein

To investigate the subcellular localization of TPD54, we generated a cell line where TPD54 was tagged at its endogenous locus with monomeric GFP (Fig. 1 and Fig. S1). GFP-TPD54 fluorescence was apparently diffuse in the cytoplasm, but was also seen at the Golgi apparatus, marked with GalT-mCherry, and on endosomes, marked by APPL1 and OCRL1. It also partially overlaps with various membrane trafficking proteins, such as clathrin light chain A and the R-SNARE VAMP2 (Fig. 1 A). A similar pattern was seen by overexpression of GFP-, mCherry-, or FLAG-tagged TPD54 in parental cells (Fig. S2). These observations suggest that TPD54 is a protein associated with membrane trafficking.

As a next step to characterizing TPD54, we investigated the binding partners of TPD54. To do so, we performed an immunoprecipitation of GFP-tagged TPD54 from HeLa cell lysates and analyzed coprecipitating proteins by mass spectrometry (Fig. 1 B). We found that two other members of the TPD52-like family, TPD52 and TPD53, were significantly enriched in the TPD54 samples versus control. TPD52, TPD53, and TPD54 have been reported to heterodimerize (Byrne et al., 1998), which suggested that this analysis was able to detect binding partners of TPD54. Among the other significant hits, we found the Rab GTPases

Rab14, Rab2a, and Rab5c. Rab14 has been identified as a regulator of the transport between the Golgi apparatus and early endosomes (Junutula et al., 2004), as well as from the Golgi apparatus to the PM (Kitt et al., 2008). Rab2a is on the ER-to-Golgi pathway (Tisdale et al., 1992), and Rab5c is found on the endocytic pathway (Bucci et al., 1995). Taken together, the results confirm that TPD54 is a protein involved in membrane trafficking.

### TPD54 is involved in multiple membrane trafficking pathways

To investigate potential functions of TPD54, we sought to identify trafficking defects caused by the loss of TPD54. Using RNAi to deplete TPD54 in HeLa cells, we first assessed the transport of cargoes from the ER to the Golgi apparatus, and from the Golgi to the PM with the RUSH (retention using selective hooks) system (Boncompain et al., 2012). Briefly, the RUSH system allows the synchronous release of a reporter (here, GFP-tagged E-cadherin with a streptavidin-binding domain) from an ER-localized hook (here, streptavidin fused to a KDEL amino acid motif) by addition of biotin. After release, EGFP–E-cadherin is transported from the ER to the PM, via the Golgi apparatus. In control cells, the reporter reached maximal intensity at the Golgi between 14 and 28 min after release and then left the Golgi for the PM (Fig. 2, A and B; and Video 1). By contrast, TPD54-depleted cells had obviously delayed kinetics (Fig. 2, A and B; and Video 2). We quantified the fluorescence of the reporter at the Golgi and expressed it as a fraction of the total cell fluorescence. The resulting data were best described by a logistic function representing ER-to-Golgi transport and a line fit to describe Golgi-to-PM (Fig. 2 B; see Materials and methods). Similar retardation of traffic was seen with three siRNAs to TPD54 (Fig. 2, C and D). This automated procedure allowed us to find the $t_{1/2}$ for ER-to-Golgi and ER-to-PM transport and also infer the Golgi transport time as the difference between these times (Fig. 2, E–G). The data suggest that TPD54-depleted cells have delayed export of E-cadherin at all stages.

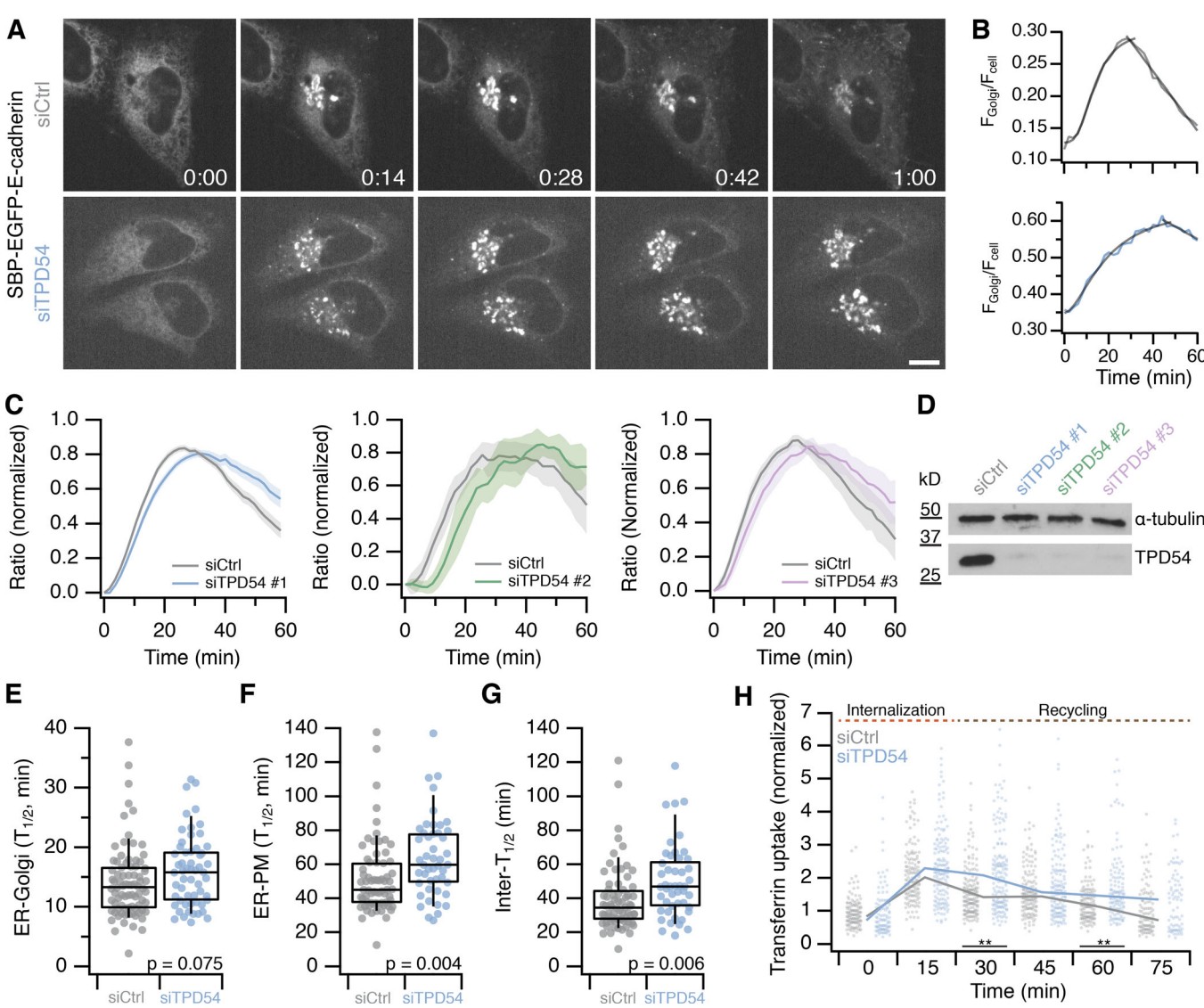

Figure 2. **TPD54-depleted cells have defective anterograde membrane traffic and cargo recycling. (A)** Still confocal images of RUSH experiments. SBP-EGFP–E-cadherin localization in control (siCtrl) and TPD54-depleted (siTPD54) HeLa cells at the indicated times (minutes and seconds) after biotin treatment. Scale bar, 10 µm. **(B)** Single cell traces of the E-cadherin fluorescence ratio of a control (gray) or TPD54-depleted (blue) cell, fitted with a logistic function and a line. **(C)** Normalized fraction of total E-cadherin fluorescence at the Golgi as a function of time in control (gray) or TPD54-depleted (colored) cells. Results from three siRNAs are shown as indicated. Line and shaded area, mean ± SEM. $n_{cell}$ = 85 (siCtrl), 62 (siTPD54 #1), $n_{exp}$ = 2; $n_{cell}$ = 23 (siCtrl), 12 (siTPD54 #2), $n_{exp}$ = 1; $n_{cell}$ = 43 (siCtrl), 20 (siTPD54 #3), $n_{exp}$ = 1. **(D)** Western blot to assess the depletion of TPD54 by RNAi for three siRNAs. The protein level of TPD54 and α-tubulin (loading control) is shown. **(E–G)** Box plots showing the $t_{1/2}$ of E-cadherin transport from ER-to-Golgi (E) and from ER-to-PM (F) in control and TPD54-depleted cells. **(G)** The difference in $t_{1/2}$ represents intra-Golgi transport. Dots represent individual cells, boxes show interquartile range, bars represents the median, and whiskers show 9th and 91st percentiles. The P values are from Student's t test with Welch's correction. $n_{cell}$ = 57–82, $n_{exp}$ = 2. **(H)** Plot showing the uptake and recycling of transferrin in control and TPD54-depleted cells. Dots represent individual cells, lines represent the median value. Wilcoxon rank test. **, P < 0.01. $n_{cell}$ = 77–160, $n_{exp}$ = 3.

We also wanted to know if TPD54 was required for endocytosis or cargo recycling since Rab5c was one of our mass spectrometry hits (Fig. 1 B). To do so, we performed a transferrin uptake and recycling assay in TPD54-depleted and control HeLa cells. The internalization of transferrin was unchanged, but recycling to the PM was slower in TPD54-depleted cells (Fig. 2 H). In these experiments, the efficiency of the depletion was checked by Western blot analysis, using α-tubulin as a loading control (Fig. 2 D).

In the RUSH experiments, we noticed that the Golgi appeared dispersed as cargo moved through it (see Fig. 2 A). Therefore,

our third functional test was to assess the distribution of the TGN using TGN46 as a marker. Depletion of TPD54 by RNAi resulted in dispersion of the TGN (Fig. 3 A). Although knockdown of TPD54 was good, as assessed by Western blot, the Golgi dispersal phenotype was mild (Fig. 3 B). Next, we knocked out the TPD54 gene in HeLa cells using CRISPR/Cas9 and recovered two independent clones that had no detectable expression of TPD54 (Fig. S3). We saw severe TGN dispersal in both clones that lacked TPD54, compared with the parental cells (Fig. 3, C and D). Importantly, normal TGN distribution could be rescued

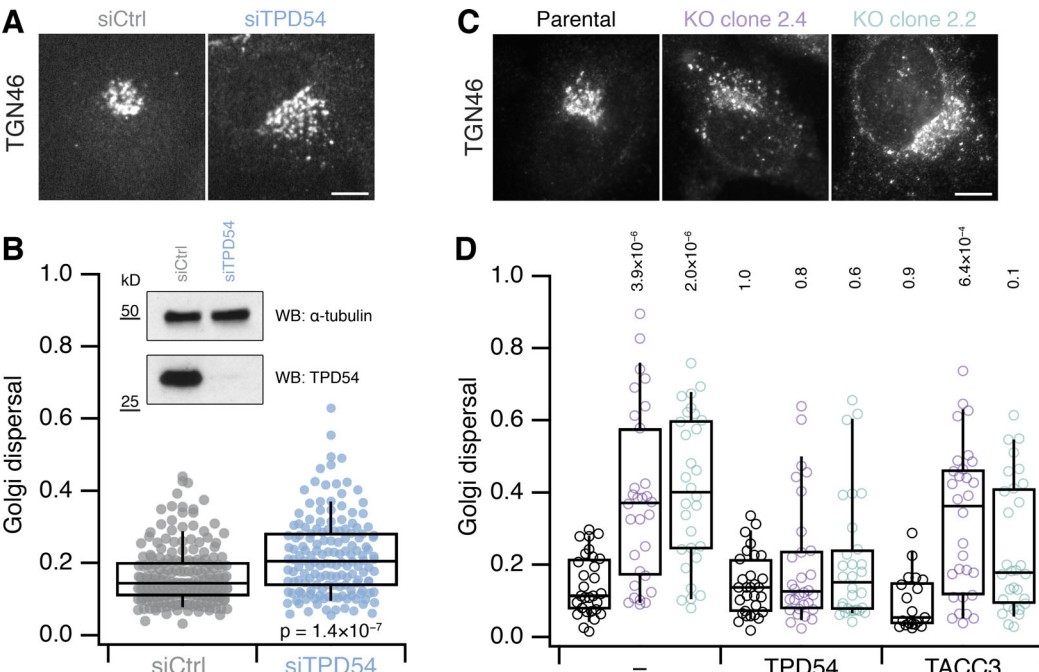

Figure 3. **Golgi dispersal in cells lacking TPD54. (A)** Micrographs of TGN46 distribution in HeLa cells treated with siCtrl (GL2) or siTPD54 (TPD54 RNAi). Scale bar, 10 µm. **(B)** Quantification of Golgi dispersal in control and TPD54-depleted cells. Golgi dispersal is the area of a convex hull of the TGN46 signal as a fraction of the total cell area. $n_{cell}$ = 155–197, $n_{exp}$ = 3. Inset, Western blot to show knockdown efficiency. **(C)** Micrographs of TGN46 distribution in parental HeLa cells and two clones with targeted disruption of the TPD54 locus (2.4 and 2.2). Scale bar, 10 µm. **(D)** Quantification of Golgi dispersal in each clone compared with parental cells. FLAG-TPD54 and FLAG-TACC3 was expressed as indicated; – indicates no reexpression. Dots represent individual cells, boxes show interquartile range, bars represents the median, and whiskers show 9th and 91st percentiles. $n_{cell}$ = 19–30, $n_{exp}$ = 3. The P values shown are from Wilcoxon rank sum test (B) or Dunnett's multicomparison test compared with control with no reexpression (D). KO, knockout.

in each clone by reexpression of FLAG-tagged TPD54, but not by expression of an unrelated protein containing a coiled-coil domain (FLAG-TACC3, Fig. 3 D). To our frustration, the Golgi dispersal phenotype in both knockout clones disappeared with repeated passaging, which might be explained by compensation for the chronic loss of TPD54 in knockout cells. These experiments confirm that the Golgi dispersal phenotype is specifically due to loss of TPD54 and is not the result of off-target action.

Together, our results suggest that TPD54 operates in several membrane trafficking pathways: (1) anterograde traffic, (2) endosomal recycling, and (3) Golgi integrity.

### Rerouting TPD54 to mitochondria changes mitochondrial morphology

Knocksideways is a standard method to remove a protein from its site of action to understand its normal function or study its binding partners (Robinson et al., 2010). To do this, proteins tagged with an FK506-binding protein (FKBP) domain can be rerouted to MitoTrap, a mitochondrially targeted FRB domain, by the addition of rapamycin. As expected, mCherry-FKBP-TPD54, but not mCherry-TPD54, was efficiently rerouted to mitochondria using this method (Fig. 4 A). The kinetics of rerouting was reasonably fast, with TPD54 appearing at mitochondria 6 s after rapamycin (Fig. 4, B and C; and Video 3). The increase in mitochondrial TPD54 was best fit by a single exponential function ($\chi^2$ = 0.43, $\tau$ = 37.98 ± 0.38 s), while the loss in cytoplasmic signal followed similar kinetics ($\tau$ = 47.52 ± 0.21 s;

Fig. 4 B). During our TPD54-rerouting experiments, we noticed that once the rerouting was complete, mitochondrial morphology became altered and the mitochondria began to aggregate (Fig. 4 C).

### Mitochondrial rerouting of TPD54 results in vesicle capture

To investigate why mitochondrial morphology became altered at later time points after rerouting TPD54, we used EM to examine the ultrastructure of mitochondria at different time points (Fig. 5). Cells expressing MitoTrap and mCherry-FKBP-TPD54 were imaged as rapamycin was applied (Fig. 5 A). They were then fixed at various times after rerouting, and the same cells were then imaged by EM. The mitochondrial TPD54 signal was partial 20 s after rapamycin; after 5 min the signal was maximal, and after 30 min mitochondrial aggregation was observed by light microscopy (Fig. 5, A and B). At the EM level, in cells where TPD54 was rerouted, mitochondria were decorated with numerous small vesicles. After 5 min or 30 min, it was clear that mitochondria had become aggregated because the vesicles had contacted more than one mitochondrial surface. We segmented the mitochondrial and vesicular profiles to analyze this effect in more detail (Fig. 5 C). The vesicles captured after TPD54 rerouting are small, homogeneous (29.9 ± 9.4 nm), and do not change size over time (Fig. 5, D and E). The number of vesicles captured per unit length of mitochondrial perimeter increases with time, and the perimeter lengths that remain undecorated decreases (Fig. 5 E). We noted significant vesicle capture at the

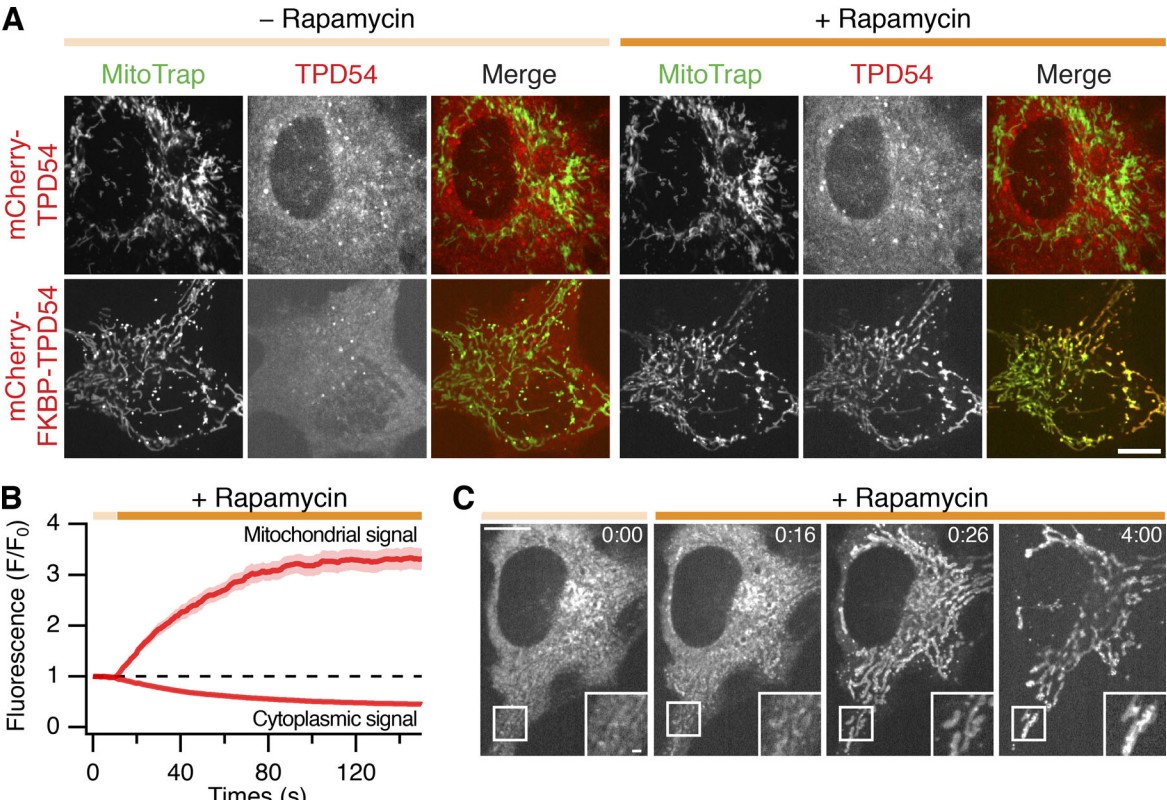

**Figure 4. TPD54 can be rerouted efficiently to mitochondria. (A)** Confocal micrographs showing the rerouting of mCherry-FKBP-TPD54, but not mCherry-TPD54, to mitochondria in cells coexpressing YFP-MitoTrap after addition of 200 nM rapamycin (orange bar). **(B)** Kinetics of mCherry-FKBP-TPD54 rerouting. The mitochondrial and cytoplasmic signal of mCherry-FKBP-TPD54 as a function of time after the addition of 200 nM rapamycin at 10 s. Line and shaded area show the mean ± SEM, $n_{cell}$ = 16. **(C)** Still images from a TPD54 rerouting video. Time, minutes and seconds (rapamycin at 0:10). Insets, 2× zoom. Scale bars, 10 µm, 1 µm (inset).

earliest time point we could study: 20 s after rapamycin. Mitochondria in control cells are essentially undecorated, with the occasional vesicle coinciding with our detection criteria, confirming that vesicle capture is a result of TPD54 rerouting to mitochondria. These experiments explain the mitochondrial aggregation and suggest that TPD54 is resident on a large population of small-size intracellular vesicles. Because of their size and lack of coat, these vesicles are unlike any formerly described class of vesicle. We refer to them as INVs.

### Visualizing INVs by light microscopy

TPD54 localizes to a small number of large puncta in cells, with the remainder being apparently cytoplasmic (Fig. 1 A). Could it be that the "cytoplasmic" TPD54 actually corresponds to a large population of small TPD54-positive vesicles (INVs) that are below the resolution limit of the microscope? In support of this idea, close inspection of our previous live-cell imaging data showed that the cytoplasmic TPD54 signal flickered as expected for mobile subresolution vesicles (for example, see Video 3 before rapamycin addition). To quantify this flickering behavior, we used the spatiotemporal variance of fluorescence in live-cell imaging videos. Indeed, the variance was over twofold greater in cells expressing GFP-TPD54 (either overexpressed or endogenous) compared with GFP, which has a uniform cytosolic distribution (Fig. 6, A and B; and Video 4). Moreover, FRAP analysis

showed slower kinetics for GFP-TPD54 compared with GFP, which rapidly recovers as a freely diffusing cytosolic protein. We could only detect a minimal freely diffusing pool of TPD54 under conditions of overexpression (summarized in Fig. S4; examples in Video 5). These results are consistent with GFP-TPD54 being absent from the cytosol, but predominantly localized on small vesicles that are below the resolution limit.

To unambiguously visualize these subresolution structures, we used stochastic optical reconstruction microscopy (STORM) to image endogenous TPD54 in GFP-TPD54 knock-in HeLa cells (Fig. 6, C–G). The reconstructed single-molecule localization microscopy images showed that the subresolution structures are, in fact, small puncta (Fig. 6, C–E). These spots had an average width of 33.6 nm, which agrees with the size of INVs observed by EM (Fig. 6 G). They were also as numerous as the vesicle capture experiment suggested, with an average density of 26.8 spots per 10 µm². Together these data indicate that TPD54 is resident on INVs and that these vesicles are not the product of the vesicle capture procedure, but are normally found in cells.

### INVs meet three criteria for functionality

The small size of INVs raised the question of whether or not they were functional. We reasoned that there are three basic criteria for a vesicle to be considered functional: it must (1) contain

**Larocque et al.**
New small transport vesicles defined by TPD54

**Journal of Cell Biology** 5

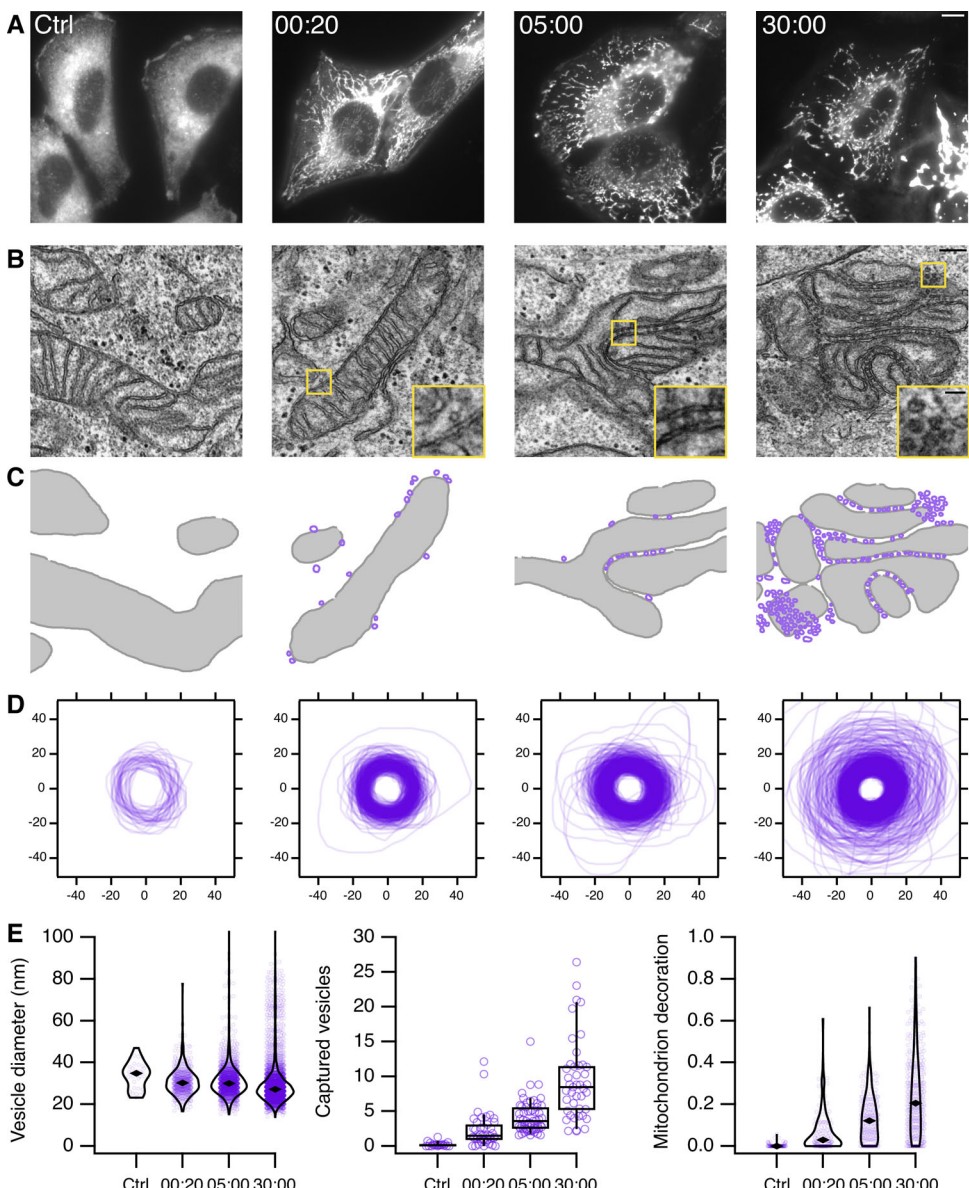

Figure 5. **Capture of small vesicles by rerouting TPD54 to mitochondria. (A)** Fluorescence microscopy images of mCherry-FKBP-TPD54 in HeLa cells. Cells expressing mCherry-FKBP-TPD54 and dark MitoTrap were fixed after no rapamycin application (Ctrl) or after 20 s, 5 min, or 30 min of rapamycin addition (200 nM). The pictured cell was then imaged by EM. Scale bar, 10 µm. **(B)** Sample electron micrographs of the cells shown in A. Insets, 3× zoom. Scale bars, 200 nm, 50 nm (insets). **(C)** Segmented view of mitochondria (gray) and vesicles (purple) in the images shown in B. **(D)** Profiles of segmented vesicles from electron micrographs. All vesicles segmented from the control dataset are shown with a random sample from the treatment groups as a comparison. The sample size is in proportion to the capture of vesicles at the mitochondria (34, 320, 594, and 1,347 for control, 20 s, 5 min, and 30 min, respectively). **(E)** Left: violin plot to show the diameter of vesicles imaged in each dataset. Spots represent individual vesicles. Marker shows the median. Center: box plot to show the number of vesicles captured per 1 µm of mitochondrial membrane. Spots show the number per micrograph in each dataset. Boxes show the interquartile range and median, and whiskers show 9th and 91st percentiles. Right: violin plot to show the fraction of mitochondrial membrane that is decorated with vesicles. Spots show individual mitochondria from the dataset. Time, minutes and seconds. See Materials and methods for details.

cargo, (2) have fusion machinery, and (3) associate with a Rab GTPase.

We first sought to identify the vesicles' cargo. To do this, we tested five model cargoes where the α chain of CD8 is fused to different peptides that bear various endocytic motifs (Kozik et al., 2010). Briefly, CD8-FANPAY, CD8-YAAL, or CD8-EAAALL has a single [F/Y]XNPX[Y/F], YXXφ, or [D/E]XXXL[L/I/M] motif, respectively (where X is any amino acid and φ is a hydrophobic amino acid). CD8-CIMPR has the tail of the cation-

independent mannose-6-phosphate receptor (CIMPR), which contains at least four endocytic motifs, including two of the dileucine type. As a control, CD8-8xA was used, which has eight alanines and no endocytic motif and therefore cannot be internalized. We examined the subcellular distribution of these cargoes in cells where TPD54 had been rerouted to mitochondria. In the control condition with no addition of rapamycin, all CD8 constructs were in endosomes or, in the case of CD8-8xA, at the PM. After rerouting, the localization of CD8-8xA, CD8-FANPAY,

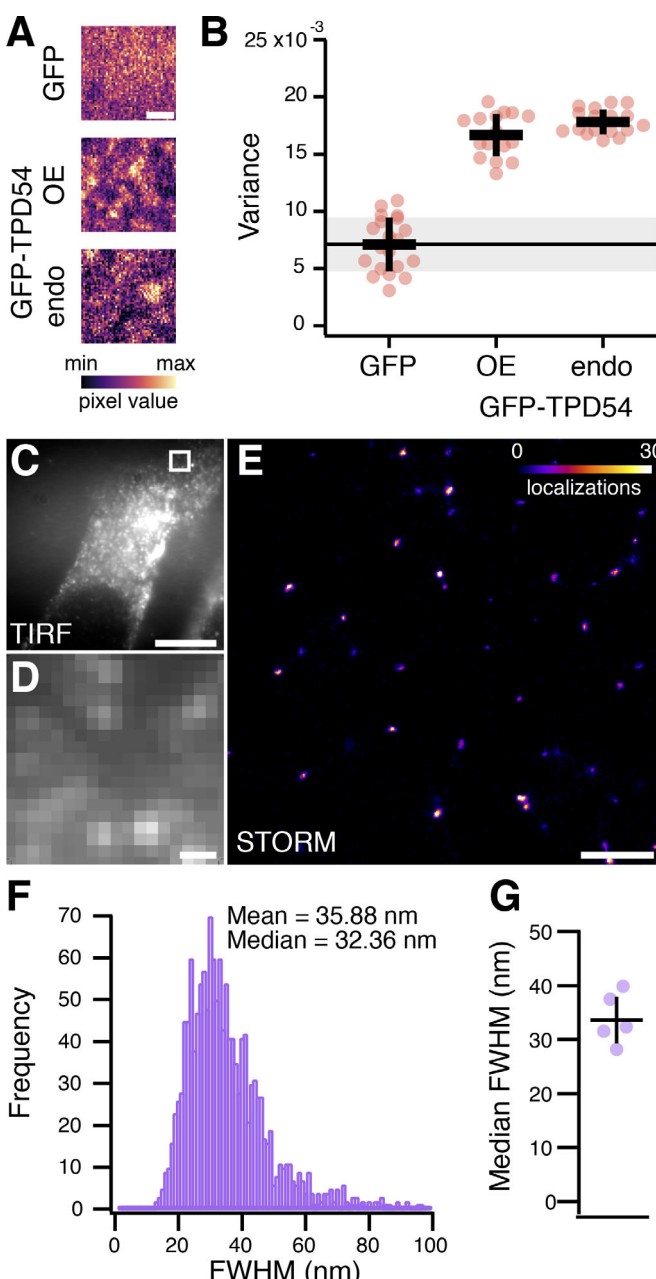

Figure 6. **Visualizing INVs by light microscopy. (A)** Single ROIs from live-cell imaging experiments with cells expressing GFP or GFP-TPD54 (overexpressed [OE]), or with knock-in GFP-TPD54 cells (endo). Images are pseudocolored to highlight subresolution structures. Scale bar, 1 μm. **(B)** Scatter dot plot to show the mean variance per pixel over time. Dots, individual cells; black bars, mean ± SD. The mean ± SD for GFP is indicated as a thin black line and gray zone. $n_{cell}$ = 16–20. **(C)** TIRF image showing the ventral surface of a typical GFP-TPD54 knock-in HeLa cell imaged by STORM. Scale bar, 10 μm. **(D)** Expanded view of the boxed region in C. Scale bar, 500 nm. **(E)** STORM image of the corresponding region shown in D. Localizations are pseudocolored as indicated; max value in image was 79. Scale bar, 500 nm. **(F)** Histogram of FWHM values of all spots in the entire localizations image for the cell shown in C. **(G)** Summary of median FWHM values. Bars, mean ± SD. $n_{cell}$ = 5, $n_{exp}$ = 3.

and CD8-YAAL was unaffected, whereas CD8-EAAALL and CD8-CIMPR were co-rerouted with TPD54 to the mitochondria (Fig. 7 A). To ensure that this co-rerouting was genuine and not a peculiarity of the model cargoes, we confirmed that endogenous CIMPR also co-rerouted with TPD54 (Fig. 7 B). This suggested that vesicles with cargo harboring a dileucine motif were preferentially captured by TPD54 rerouting.

Having captured specific cargo that is present in INVs at steady-state, we next tested if INVs were actively trafficking cargo. Receptors containing dileucine endocytic motifs are internalized at the PM and then recycled via either recycling endosomes or the Golgi apparatus. We therefore labeled CD8-EAAALL at the surface with Alexa Fluor 488–conjugated anti-CD8 antibodies, allowed internalization and trafficking to proceed, and, at different time points, performed mitochondrial vesicle capture via TPD54 rerouting (Fig. 7 C). Capture of surface-labeled CD8-EAAALL occurred at time points >60 min after internalization (Fig. 7 D). These experiments indicate that dileucine motif–containing receptors transit via INVs, which can be captured on mitochondria by TPD54 rerouting, but only at late time points after internalization. The time course of capture is consistent with recycling of endocytic cargo from the Golgi apparatus.

The second criterion for vesicle functionality is whether the vesicles contain the machinery for fusion. Accordingly, we tested for co-rerouting of endogenous SNAREs in our vesicle capture assay. Generally, vesicle-resident R-SNAREs, but not target membrane-resident Q-SNAREs, were co-rerouted with TPD54 to mitochondria. We found co-rerouting of the R-SNAREs VAMP2, VAMP3, VAMP7, and VAMP8, but not the Q-SNAREs STX6, STX7, STX8, STX10, or STX16 (Fig. 8). There was some evidence of selectivity with the localization of the R-SNARE VAMP4 being unaffected by TPD54 rerouting. Moreover, the presence of different SNAREs suggests that although the captured INVs appear morphologically homogeneous, they are likely to be a crowd of different vesicle identities.

What are the identities of the vesicles captured by TPD54 rerouting? To answer this question, we screened 43 GFP-tagged Rab GTPases for co-rerouting with mCherry-FKBP-TPD54 to dark MitoTrap. The collection of GFP-Rabs tested covers a range of membrane trafficking pathways (Yoshimura et al., 2007; Zhen and Stenmark, 2015; Wandinger-Ness and Zerial, 2014). The results of the screen are presented in Fig. 9, with examples of positive and negative hits shown in Fig. S5 A. This screen confirmed that INVs meet the third criterion for functionality: being associated with specific Rabs.

**INVs have a heterogeneous complement of Rab GTPases**

In the vesicle capture screen, significant co-rerouting was detected for 16 out of 43 Rabs. These positive hits were Rab30, Rab25, Rab26, Rab45, Rab14, Rab11a, Rab12, Rab1a, Rab43, Rab1b, Rab10, Rab33b, Rab19, Rab33a, Rab37, and Rab2a (listed by descending effect size; Fig. 9 C). Some evidence for co-rerouting of Rab38, Rab5c, and Rab35 was seen, although in any individual trial no clear difference was observed. The localization of the other 24 Rabs was unaffected by rerouting of TPD54 and was indistinguishable from GFP. Rab30 was the most efficiently

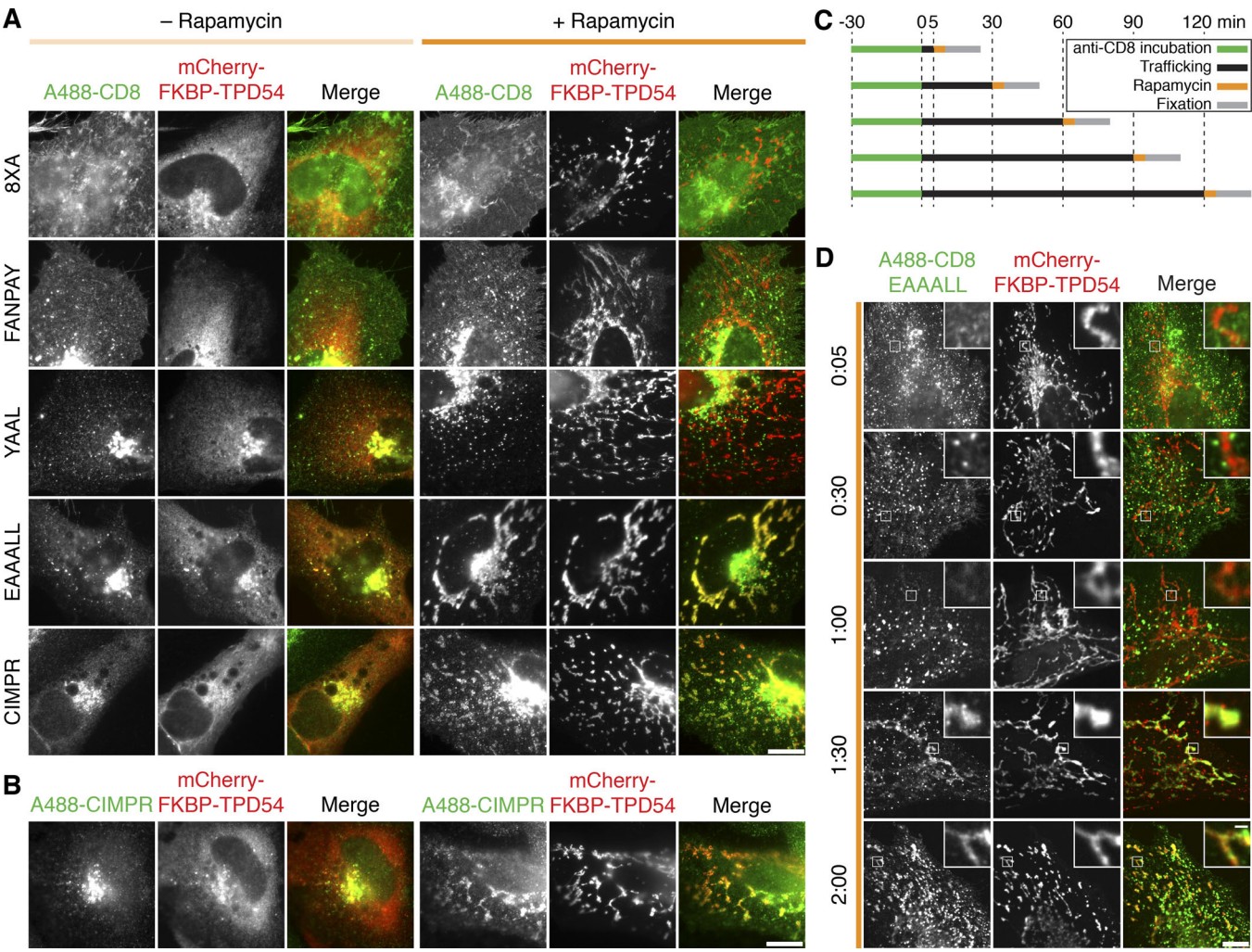

Figure 7. **TPD54 co-reroutes dileucine motif-containing receptors only. (A)** Representative widefield micrographs of cells coexpressing mCherry-FKBP-TPD54, dark MitoTrap, and the indicated CD8 construct. Rerouting was induced by 200 nM rapamycin. Cells were fixed, permeabilized, and stained for total CD8. **(B)** Representative widefield micrographs showing co-rerouting of endogenous CIMPR detected by immunofluorescence with rerouting of mCherry-FKBP-TPD54 to dark MitoTrap by addition of 200 nM rapamycin. **(C)** Pulse label and timed vesicle capture experiments. Cells expressing CD8-EAAALL were surface labeled with Alexa Fluor 488–conjugated anti-CD8 antibodies for 30 min, then incubated at 37°C for the indicated time (minutes), treated with 200 nM rapamycin for 5 min, and fixed. **(D)** Representative widefield micrographs from a pulse label and timed vesicle capture experiment. Inset, 5× zoom. Scale bars, 10 µm, 1 µm (insets). Time, hours and minutes.

co-rerouted Rab, with the post-rerouting signal being 2.5-fold higher than before TPD54 rerouting (Video 6). The smallest effect that we could reliably detect was Rab2a, with a 1.4-fold increase. In the case of Rab1a, we confirmed that co-rerouting was also seen with the endogenous protein (Fig. S5 B).

If the relocalization of Rabs observed in the screen was the result of co-rerouting with TPD54, a correlation between the extent of rerouting for a Rab and TPD54 is predicted (Fig. 9 D). This was broadly true, with a positive correlation observed for almost all of our positive hits, and a low, flat relationship for negative Rabs. Rab14 was an exception. Here, the relationship was high and flat; Rab14 rerouting was maximal even after modest TPD54 rerouting. This result is consistent with there being a very limited pool of Rab14-positive vesicles, all of which are TPD54 positive.

We next performed a test of reciprocality by asking if mCherry-TPD54 was co-rerouted to mitochondria when a GFP-

FKBP-Rab was rerouted to dark MitoTrap using 200 nM rapamycin. We tested two positive hits from our screen, Rab11a and Rab25, as well as a negative, Rab7a (Fig. S5 C). Rerouting of either Rab11a or Rab25 caused co-rerouting of TPD54, while rerouting Rab7a to mitochondria had no effect on TPD54 localization. Interestingly, we noticed that when Rab11a or Rab25 was rerouted, there was still a number of TPD54-positive structures, presumably associated with other Rabs, that were not rerouted (Fig. S5 C). TPD54 rerouting tended to give a more complete removal of Rab-positive structures from the cytoplasm (Fig. S5 A). This observation supports the idea that TPD54 defines a class of vesicle that each bears a Rab from a large subset of Rab GTPases. The collective heterogeneity of Rabs and R-SNAREs on INVs suggests that this class of transport vesicle has diverse origins and varied destinations. The summary in Fig. 10 shows the results from the Rab screen on a cellular map

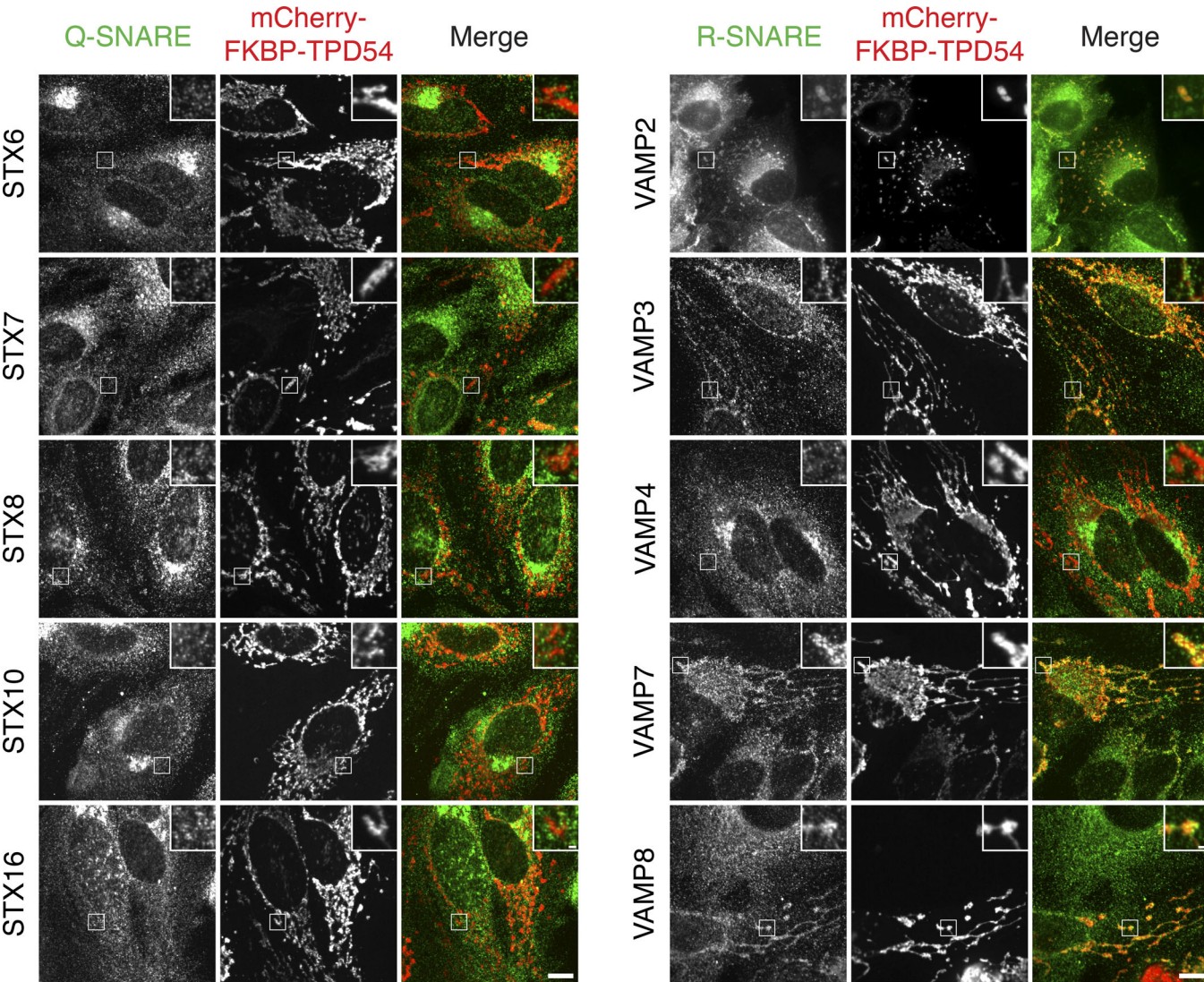

Figure 8. **Co-rerouting of R-SNARES, but not Q-SNARES, with TPD54.** Representative confocal micrographs showing the co-rerouting of SNAREs as indicated after TPD54 rerouting to mitochondria. SNAREs were detected by immunofluorescence, with the exception of VAMP2, which is coexpressed as GFP-VAMP2 (widefield image). Insets, 3× zoom. Scale bars, 10 µm, 1 µm (insets).

of intracellular trafficking pathways. Several pathways are "ruled out" due to their governance by Rabs which were negative in our screen. Positive hits coincide with anterograde or recycling pathways and with Golgi transport, which is in agreement with our functional data on the function of TPD54. Our proposed model for INVs, therefore, is that they represent a generic class of transport vesicle that is found throughout much of the membrane traffic network.

## Discussion

This study shows that TPD54, an abundant protein in mammalian cells, is found on numerous small vesicles throughout the cell. These vesicles—INVs—are functional since they have cargo, fusion machinery, and Rabs. The heterogeneity of Rabs and SNAREs suggests that INVs are generic transport carriers that mediate transport between diverse originating membranes

and various destinations. Accordingly, we saw that loss of TPD54 interferes with several membrane traffic steps: anterograde traffic, recycling, and Golgi integrity.

We found INVs serendipitously. Using a knocksideways-based system, when TPD54 was rerouted to mitochondria, we saw that it was associated with small vesicles. There have been previous reports of vesicle capture at mitochondria following knocksideways of gadkin (Hirst et al., 2015), or by ectopic mitochondrial expression of Golgins (Wong and Munro, 2014). However, the vesicles captured by rerouting TPD54 are smaller: 29.9 ± 9.4 nm in diameter. Vesicles of a similar size distribution were also observed under normal conditions by STORM imaging. Few types of vesicles are this small. For example, clathrin-coated vesicles are 50–100 nm, COPII-coated vesicles are 60–70 nm, and intra-Golgi transport vesicles are 70–90 nm (Vigers et al., 1986; Balch et al., 1994; Orci et al., 2000). Synaptic vesicles are a good size match at 33–38 nm, but they are restricted to

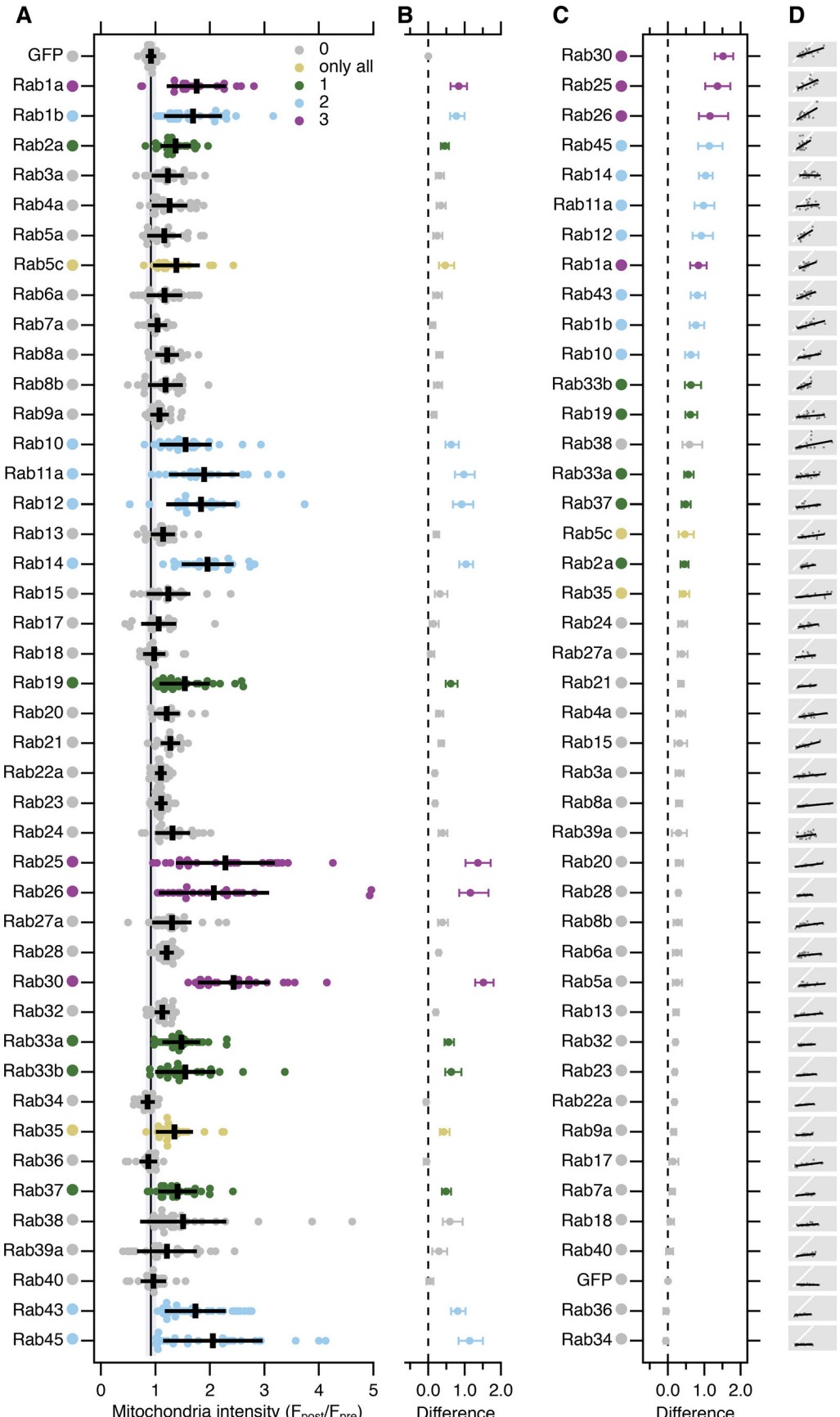

Figure 9.  **A screen to identify Rab GTPases that are associated with TPD54. (A)** Quantification of the change in mitochondrial fluorescence intensity of GFP or GFP-Rabs 2 min after rerouting of mCherry-FKBP-TPD54 to dark MitoTrap with 200 nM rapamycin. Multiple independent experiments were completed (dots) across three independent trials. Black bars, mean ± SD. The mean ± SD for GFP (control) is also shown as a black line and gray zone, down the plot. Dunnett's post-hoc test was done for each trial using GFP as a control. Colors indicate if $P < 0.05$ in one, two, or three trials, or only when all the data were pooled. $n_{cell}$ = 17–36, $n_{exp}$ = 3. **(B)** Effect size and bootstrap 95% confidence interval of the data in A. **(C)** The plot in B is reordered to show Rabs ranked in order of highest to lowest effect size. **(D)** Small multiple plots show the correlation between the mCherry-FKBP-TPD54 rerouting and GFP-Rabs co-rerouting (gray dots), a line fit to the data (black), and a y = x correlation (white).

neurons (Harris and Sultan, 1995). In nonneuronal cells, the closest-sized vesicles are intralumenal vesicles, which range from 20 to 100 nm (Edgar et al., 2014; Raposo and Stoorvogel, 2013). However, their inaccessibility and opposite orientation makes intralumenal vesicles an unlikely candidate for the vesicles captured by TPD54 rerouting. Our interpretation is that INVs are an overlooked class of intracellular vesicle. With no coat to distinguish them and with an unimposing size, these inconspicuous vesicles seem to have evaded study until now.

Is it possible that TPD54 is on a wider variety of vesicles, but that rerouting only captures the smallest of all TPD54-positive vesicles? One could imagine that smaller vesicles are captured more efficiently than larger ones. However, larger vesicles and even Golgi cisternae can be captured by mitochondria under different experimental conditions (Hirst et al., 2015; Wong and Munro, 2014; Shin et al., 2017; Dunlop et al., 2017), suggesting that TPD54 is predominantly localized to these small vesicles and that this is the reason why they are captured more efficiently.

Moreover, it is unlikely that the captured vesicles are the result of vesicularization of larger membranes since we saw the capture of 30 nm vesicles after only 20 s of TPD54 rerouting and no further change in size of captured vesicles at longer time points. In addition, super-resolution imaging also showed only small puncta.

INVs appear to be real, functional transport carriers because the vesicles we captured had cargo, fusion machinery, and Rabs. There was selectivity in their cargo, which all featured dileucine-type endocytic motifs (Bonifacino and Traub, 2003), and in their R-SNARE complement. The presence of SNAREs in INVs could be determined by their dileucine sorting motifs, although we saw no co-rerouting of VAMP4, which has one such motif (Peden et al., 2001; Gordon et al., 2009). How the cargo and SNAREs are sorted into INVs is an interesting question for future investigation, as is the broader question of how they bud. Our rerouting assay reports which proteins were passengers in INVs, but does not tell us which of these, if any, TPD54 binds

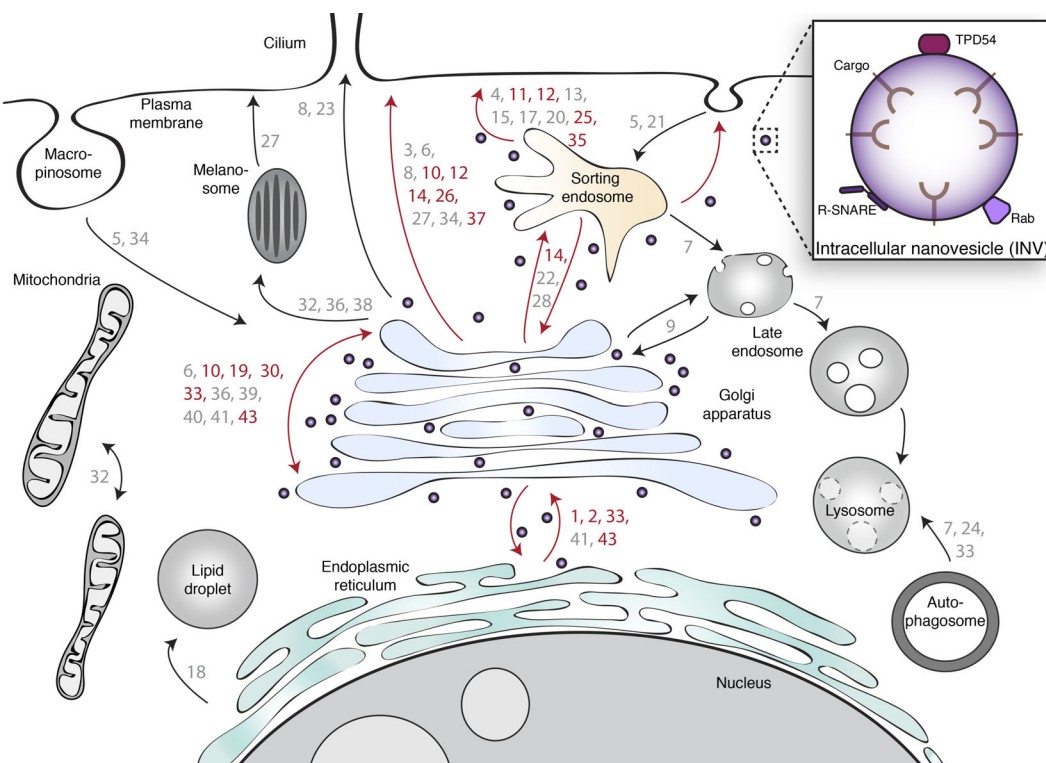

Figure 10.  **Cellular localization of INVs.** Schematic diagram showing the cellular pathways on which the Rab GTPases operate. Rabs are represented by their number. Red and gray numbers indicate positive and negative Rab hits, respectively. Red and black arrows indicate pathways that involve a Rab that is a positive hit or where only negative hits were found, respectively. INVs are shown as purple vesicles peppered throughout most, but not all, of the network. The inset summarizes the three types of protein found on INVs that were tested in the present study.

directly. In the case of Rab GTPases, we found that Rab2a, Rab5c, and Rab14 co-immunoprecipitate with TPD54. Rab2a and Rab14 were positive hits in our screen, while Rab5c was a borderline hit. It is possible that TPD54 binds to these Rabs as an effector. Previous work indicates that TPD52-like proteins can bind Rabs. Rab5c was identified as a binding partner for TPD52 (Shahheydari et al., 2014), and an indirect association of Rab5, Rab6, and Rab9 with TPD52 has also been reported (Zhang et al., 2007), although Rab5a, Rab6, and Rab9 were all negative in our screen using TPD54. Recently, MitoID was used to identify Rab interactors, and TPD54 was not a top ranked hit for any of the Rabs tested (Gillingham et al., 2019). Considering this and the observation that at least 16 Rab GTPases from three different Rab subgroups co-reroute with TPD54, it would seem unlikely that TPD54 binds them all (Klöpper et al., 2012). We note that OCRL1 binds to multiple Rabs (six Rabs from four different subgroups) via a single domain of 123 residues (Hou et al., 2011). Our working model is that TPD54 can bind directly to INVs; interactions with Rabs are possible but not necessary for INV localization.

The generation of small-size vesicles makes a lot of sense for a transport network. Just as bicycle couriers are the fastest physical delivery agents in crowded cities, small vesicles should be able to rapidly access the most congested parts of cells to deliver their cargo. For example, when cargo is recycled to the PM, much attention has been focused on large tubules that emerge from the sorting endosome (Geuze et al., 1983). However, analysis of the delivery of receptors at the cell surface indicates that the final carrier is a small-size vesicle rather than a large tubule (Xu et al., 2011; Shen et al., 2014). This raises the possibility that INVs may bud from large secondary carriers as well as from the primary originating organelle. While being small has its advantages, a major disadvantage is that the capacity of INVs is restricted. Small vesicles can actually carry a surprising amount of cargo (Takamori et al., 2006; Martins Ratamero and Royle, 2019 Preprint), but they are presumably restricted for the carriage of large cargo: i.e., those with bulky extracellular domains or large ligands (Martins Ratamero and Royle, 2019 Preprint), although not necessarily (McCaughey et al., 2019). Other classes of vesicle exhibit size adaptability. Clathrin-coated vesicles can vary in size (Miller et al., 2015), and very large clathrin-coated vesicles can form in some preparations (Perry and Gilbert, 1979). Intralumenal vesicles become larger after internalization of EGF, suggesting that cargo can influence vesicle size (Edgar et al., 2014). Whether INVs show similar adaptability as other vesicle classes, and under what circumstances, remains to be tested.

## Materials and methods
### Molecular biology
GFP-TPD54, mCherry-TPD54, and FLAG-TPD54 were made by amplifying TPD54 by PCR from human tumor protein D54 (IMAGE clone, 3446037) and inserted into either pEGFP-C1, pmCherry-C1, or pFLAG-C1 via XhoI-MfeI. GFP-FKBP-TPD54 was made by ligating a XhoI-BamHI fragment from mCherry-TPD54 into pEGFP-FKBP-C1. This plasmid was converted to mCherry-FKBP-TPD54 by cutting at BamHI-MfeI and inserting into pmCherry-C1.

YFP-MitoTrap and the CD8 chimeras were gifts from Scottie Robinson (University of Cambridge, Cambridge, UK; Addgene, #46942), and mCherry-MitoTrap was previously described (Cheeseman et al., 2013; Addgene, #59352). The dark MitoTrap (pMito-dCherry-FRB) has a K70N mutation in mCherry (Wood et al., 2017).

The GFP-Rab constructs were a gift from Francis Barr (University of Oxford, Oxford, UK), except for GFP-Rab1a and GFP-Rab5c, which were made by amplifying human Rab1a or Rab5c (Rab1a: Addgene, #46776; Rab5c: GeneArt synthesis) by PCR and inserting the genes in pEGFP-C1 via SacI-KpnI. GFP-FKBP-Rab11 and GFP-FKBP-Rab25 were a gift from Patrick Caswell (University of Manchester, Manchester, UK), and GFP-FKBP-Rab7a was made by inserting Rab7a in pEGFP-FKBP-C1 via SacI-SalI.

The plasmid to express mCherry-OCRL1 was a gift from Martin Lowe (University of Manchester). GalT-mCherry was made by cutting GalT via BamHI and MfeI from GalT-CFP (gift from Ben Nichols, MRC Laboratory of Molecular Biology, Cambridge, UK) and inserting into pmCherry-N1. GFP-VAMP2 and mCherry-VAMP2 were made by amplifying VAMP2 from synaptopHluorin (gift from James Rothman, Yale School of Medicine, New Haven, CT) and inserted into pEGFP-C1 or mRFP-C1 via HindIII and EcoRI. FLAG-TACC3 was made by amplifying TACC3 (IMAGE clone, 6148176; GenBank accession no. BC106071) by PCR and inserting into pFLAG-C1 via XmaI and MluI. mCherry-LCa was made by pasting LCa into pmCherry-C1 via BglII and EcoRI. SBP-EGFP-E-cadherin and APPL1-mCherry were obtained from Addgene (#65292 and #27683, respectively).

### Cell culture
HeLa cells (Health Protection Agency/European Collection of Authenticated Cell Cultures, #93021013) were maintained in DMEM supplemented with 10% FBS and 100 U/ml penicillin/streptomycin at 37°C and 5% $CO_2$. RNAi was done by transfecting 100 nM siRNA (TPD54 #1, 5′-GUCCUACCUGUUACGCAAU-3′; TPD54 #2, 5′-CUCACGUUUGUAGAUGAAA-3′; TPD54 #3, 5′-CAUGUUAGCCCAUCAGAAU-3′; siCtrl, 5′-CGTACGCGGAATACTTCGA-3′) with Lipofectamine 2000 (Thermo Fisher Scientific) according to the manufacturer's protocol. For DNA plasmids, cells were transfected with a total of 300 ng DNA per well of a 4-well, 3.5-cm dish using 0.75 μl Genejuice (Merck Millipore) following the manufacturer's protocol. Cells were imaged 1 d after DNA transfection and 2 d after siRNA transfection (RUSH and recycling). For two rounds of RNAi (Golgi integrity), HeLa cells were transfected with the TPD54-targeting siRNA for 48 h and transfected again with the siRNA for an additional 72 h.

The GFP-TPD54 CRISPR knock-in HeLa cell line was generated by transfecting the Cas9n D10A nickase plasmid containing the guide pairs (guide 1, 5′-ACCGCTGTCGCGGGCGCTAT-3′; guide 2, 5′-GCCCGAACATGGACTCCGC-3′) and the repair template. 9 d after transfection, GFP-positive cells were selected by FACS and isolated. Clones were validated by Western blotting and genome sequencing (sequencing primers: 5′-CAGTTTCGGCCTATCAGGTTGAGTC-3′; 5′-GAACCACACCTCGGAACGGTC-3′;

5′-CAGCTTGTGCCCCAGGATGTTG-3′; 5′-CAACTACAAGACCCG CGCCGAG-3′).

The TPD54 knockout HeLa cell lines were generated by transfecting the Cas9 plasmid containing one of the three guide pairs (guide 1, 5′-CACCGTCGCGGATTACGAAACGCCG-3′; guide 2, 5′-CACCGTTTCGTAATCCGCGATGCGA-3′; guide 3, 5′-CAC CGACCGCTGTCGCGGGCGCTAT-3′). The transfected cells were selected with 1 mg/ml puromycin 24 h after transfection. Clones were isolated and validated by Western blot and sequencing genomic DNA.

## Biochemistry

For Western blot analysis, the antibodies used were rabbit anti-TPD54 (Dundee Cell Products) 1:1,000, mouse anti–α-tubulin (Abcam, ab7291) 1:10,000, mouse anti-GFP clones 7.1 and 13.1 (Roche, 118144600010) 1:1,000, and mouse anti-clathrin heavy chain TD.1 (hybridoma) 1:1,000.

For immunoprecipitations, two 10-cm dishes of confluent HeLa cells expressing either GFP or GFP-TPD54 were used for each condition (10 μg DNA transfected per 10-cm dish). Cells were lysed in lysis buffer (10 mM Tris-HCl, pH 7.5, 150 mM NaCl, 0.5 mM EDTA, 0.5% NP-40, and protease inhibitors; Roche). The lysate was then incubated for 1 h with GFP-Trap beads (ChromoTek) and washed once with exchange buffer (10 mM Tris-HCl, pH 7.5, 150 mM NaCl, and 0.5 mM EDTA) and three times with wash buffer (10 mM Tris-HCl, pH 7.5, 500 mM NaCl, and 0.5 mM EDTA). The immuno-precipitations were run on a 4–15% polyacrylamide gel until they were 1 cm into the gel. The columns were then cut and sent for mass spectrometry analysis to the FingerPrints Proteomics Facility (University of Dundee, Dundee, UK). Protein scores from four experiments were used to make the volcano plot in IgorPro.

## Immunofluorescence

HeLa cells grown on coverslips were fixed at RT with 3% paraformaldehyde, 4% sucrose in PBS for 15 min, and permeabilized at RT in 0.1% saponin for 10 min (for all staining, unless stated otherwise). For LAMP1 staining, cells were fixed and permeabilized with ice-cold methanol at –20°C for 10 min. For CD8, TGN46, EEA1, Rab1a, CIMPR, and FLAG staining, cells were fixed in 3% paraformaldehyde, 4% sucrose in PBS for 15 min, and permeabilized at RT in 0.5% Triton-X100 in PBS for 10 min. Cells were then blocked in 3% BSA in PBS for 1 h. Cells were incubated for 1 h at RT, with primary antibodies used as follows: mouse anti-EEA1 (BD Biosciences, 610457) 1 μg/ml, rabbit anti-LAMP1 (Cell Signaling, 9091) 1:200, sheep anti-TGN46 (AbD Serotec, AHP500G) 1.25 μg/ml, mouse anti-CD8 (Bio-Rad, MCA1226GA) 10 μg/ml, rabbit anti-Rab1a (Cell Signaling, D3X9S) 0.8 μg/ml, rabbit anti-CIMPR (Thermo Fisher Scientific, PA3-850) 1:500, and mouse anti-FLAG M2 (Sigma, F1804) 1 μg/ml. Anti-SNARE antibodies were a gift from Andrew Peden (University of Sheffield, Sheffield, UK): rabbit anti-VAMP3 (1:200), rabbit anti-VAMP4 (1:500), rabbit anti-VAMP7 (1:50), rabbit anti-VAMP8 (1:100), rabbit anti-STX6 (1:200), rabbit anti-STX7 (1:400), rabbit anti-STX8 (1:100), rabbit anti-STX10 (1:50), and mouse anti-STX16 (1:200), described in

Gordon et al. (2010). Cells were washed three times with PBS for 5 min and incubated for 1 h at RT with Alexa Fluor (Invitrogen) secondary antibodies. To co-reroute the CD8-EAAALL chimera after timed incubation with anti-CD8 antibodies, cells were labeled with 10 μg/ml Alexa Fluor 488–conjugated anti-CD8 antibodies (AbD Serotec, MCA1226A488) at 4°C for 30 min. Cells were then incubated at 37°C in warm growth medium for the indicated time points. Rerouting was done by adding 200 nM rapamycin at 37°C. Cells were fixed and mounted after 5 min, as described above.

For STORM, the sample was prepared as described in Jimenez et al. (2019). GFP-TPD54 knock-in CRISPR HeLa cells were fixed for 10 min at 37°C in prewarmed fixation buffer (4% formaldehyde, 4% sucrose, 80 mM Pipes, 5 mM EGTA, and 2 mM MgCl$_2$, pH 6.8). Cells were then washed three times with PBS, permeabilized, and blocked at the same time for 1 h at RT in blocking buffer (0.22% gelatin from bovine skin type B, 0.1% Triton-X100, and PBS). Anti-GFP polyclonal primary antibodies (Invitrogen, A-11122) were diluted in blocking solution (10 μg/ml) and applied for 2.5 h at RT. The cells were washed three times for 10 min at RT in blocking solution with gentle agitation and incubated with Alexa Fluor 647–conjugated secondary antibodies for 1 h at RT. Cells were washed three times for 10 min in blocking solution with agitation at RT and brought directly to the microscope for STORM.

For the transferrin assay, HeLa cells grown on coverslips were serum starved for 30 min at 37°C, and then incubated at 4°C for 30 min with 25 μg/ml of Alexa Fluor 488–conjugated transferrin (Thermo Fisher Scientific, 11550756). The coverslips were then dipped in distilled H$_2$O, placed in warm growth medium, and incubated at 37°C for 5–75 min to allow internalization and recycling before fixation.

## Confocal microscopy

Cells were grown in 4-well, glass-bottom, 3.5-cm dishes (Greiner Bio-One), and media were exchanged for Leibovitz L-15 CO$_2$-independent medium for imaging at 37°C on a spinning disc confocal system (Ultraview Vox, PerkinElmer) with a 100× 1.4 NA oil-immersion objective. Images were captured using an ORCA-R2 digital charge-coupled device camera (Hamamatsu) following excitation with 488-nm and 561-nm lasers. For the RUSH assay, SBP-EGFP–E-cadherin was released from the ER by adding a final concentration of 40 μM D-Biotin (Sigma) in Leibovitz L-15 medium. Images were captured at an interval of 2 min. Rerouting of mCherry-FKBP-TPD54 to the mitochondria (dark MitoTrap) was induced by addition of 200 nM rapamycin (Alfa Aesar). Rerouting kinetics experiments were measured by recording videos of 150 s (1 frame per second), where rapamycin is added after 10 s. The kinetics of mCherry-FKBP-TPD54 rerouting to mitochondria was similar in cells with or without depletion of endogenous TPD54. For the Rab GTPase co-rerouting experiments, an image before rapamycin and an image 2 min after rapamycin were taken of live cells. For the FRAP experiment, a cytoplasmic region of 6.69 μm × 10.76 μm was bleached using a 488-nm laser for five cycles of 100 ms. Images were captured at the highest frame rate possible (0.1775 s).

## Correlative light-EM

Following transfection, cells were plated onto gridded dishes (P35G-1.5-14-CGRD, MatTek). Light microscopy was done using a Nikon Ti epifluorescence microscope, a heated chamber (OKOlab), and a CoolSnap MYO camera (Photometrics) using NIS-Elements AR software. During imaging, cells were kept at 37°C in Leibovitz L-15 $CO_2$-independent medium supplemented with 10% FBS. Transfected cells were found, and the grid coordinate containing the cell of interest was recorded at low magnification. Live-cell imaging was done on a cell-by-cell basis at 100×. During imaging, 200 nM (final concentration) rapamycin was added for variable times before the cells were fixed in 3% glutaraldehyde and 0.5% paraformaldehyde in 0.05 M phosphate buffer, pH 7.4, for 1 h. Aldehydes were quenched in 50 mM glycine solution and thoroughly washed in $H_2O$. Cells were postfixed in 1% osmium tetroxide and 1.5% potassium ferrocyanide for 1 h and then in 1% tannic acid for 45 min to enhance membrane contrast. Cells were rinsed in 1% sodium sulfate and then twice in $H_2O$ before being dehydrated in grade-series ethanol and embedded in EPON resin (TAAB). The coverslip was removed from the polymerized resin, and the grid was used to relocate the cell of interest. The block of resin containing the cell of interest was then trimmed with a glass knife, and serial 70-nm ultrathin sections were taken using a diamond knife on an EM UC7 (Leica Microsystems) and collected on formvar-coated hexagonal 100 mesh grids (EM Resolutions). Sections were poststained with Reynolds lead citrate for 5 min. Electron micrographs were recorded using a JEOL 1400 transmission EM operating at 100 kV using iTEM software.

## STORM

STORM was performed using a custom-built total internal reflection fluorescence (TIRF) widefield microscope with enhanced stability (Huang et al., 2016) with a 100× 1.49 NA oil-immersion objective and Andor iXon X3 camera controlled by µManager. Cells were placed between two coverslips, and STORM buffer (50 mM Tris, pH 8, 10 mM NaCl, 10% glucose, 50 mM cysteamine [Sigma, 60–23–1], 0.5 mg/ml glucose oxydase [Sigma, G2133], and 40 µg/ml catalase [Sigma, C40]) was added to the cells (Jimenez et al., 2019). A TIRF image was acquired, followed by a STORM acquisition. 12,000 images (60-ms exposure time) were acquired with the 647-nm laser. Fluorophores were reactivated during imaging by increasing illumination with a 405-nm laser every 20 frames. The STORM images were reconstructed using the Fiji plugin GDSC single-molecule localization microscopy (Schindelin et al., 2012).

## Image analysis

For analysis of RUSH videos, a region of interest (ROI) was drawn around the cell and around the Golgi apparatus in Fiji. The area, mean pixel intensity, and integrated density were measured from these ROIs to get fluorescence intensity ratios. Data were processed in IgorPro using custom scripts. Briefly, a logistic function (Eq. 1) was fitted to the data using the start of the video and two frames after the maximum value as limits for the fit:

$$f(x) = y_0 + \frac{(y_{max} - y_0)}{1 + \left(\frac{x_{1/2}}{x}\right)^n}. \tag{1}$$

The value for $x_{1/2}$ in minutes was used for the $t_{1/2}$ for ER-to-Golgi. The corresponding $y$ value for $x_{1/2}$ was used to find the $t_{1/2}$ for ER-to-PM. A line of best fit from $y_{max}$ to the end of the trace was found using $f(x) = a + bx$, and the y value corresponding to $x_{1/2}$ was found. The Golgi transit time is taken from the difference between the two $t_{1/2}$ values.

Rerouting kinetics was quantified by averaging the pixel intensity of 10 ROIs of 10 × 10 pixels on mitochondria and four ROIs of varying size in the cytoplasm per cell throughout the duration of the videos. The images were corrected for photobleaching using the simple ratio method before measuring pixel intensity for all ROIs. The intensity and time data were fed into IgorPro, and a series of custom-written functions processed the data.

For vesicle capture analysis, electron micrographs were manually segmented in IMOD software using a stylus by a scientist blind to the experimental conditions. The coordinates corresponding to contours and objects were fed into IgorPro using the output from model2point. All coordinates were scaled from pixels to real-world values, and the vesicle diameters were calculated using the average of the polar coordinates around the vesicle center for each vesicle, along with other parameters. As a metric for vesicle capture, the length of the mitochondrial perimeter was measured and used to express the vesicle abundance per image (vesicles per 1 µm). The intersection of an area corresponding to the vesicles, dilated by 15 nm and the mitochondrial perimeter, was used to express the fraction of mitochondrial perimeter that was decorated with vesicles. Analysis of "flicker" in live-cell videos was done using 50 × 50-pixel excerpts of 30 frames from live-cell imaging captured at 0.1775 s per frame. Each frame was first normalized to the mean pixel intensity for that frame, and then the variance per pixel over time was calculated, resulting in a 50 × 50 matrix of variances. The mean of this matrix is presented as the metric of flicker for that cell.

For unbiased estimation of INVs in STORM images, the $xy$ coordinates of spots in an image of localizations were logged using "Find Maxima" in Fiji (prominence >5). The localizations image and the coordinates were brought into IgorPro, and a 2D Gaussian function (Eq. 2) was fitted to a 41 × 41-pixel image centered on each coordinate:

$$f(x,y) = z_0 + A\exp\left[-\left(\frac{(x - x_0)^2}{2\sigma_x^2} + \frac{(y - y_0)^2}{2\sigma_y^2}\right)\right]. \tag{2}$$

The spot width was taken as the full-width half-maximum (FWHM; Eq. 3):

$$FWHM = 2\sqrt{2\ln2}\,\sigma_y. \tag{3}$$

For the Rab screen, co-rerouting of Rab GTPases was quantified by averaging for each cell the pixel intensity in the green channel in 10 ROIs of 10 × 10 pixels on the mitochondria, before and after rapamycin. This mitochondrial intensity ratio ($F_{post}/F_{pre}$) for every Rab was compared with the ratio of GFP in TPD54-rerouted cells. Estimation statistics were used to

generate the difference plot shown in Fig. 9 B. The mean difference is shown together with bias-corrected and accelerated 95% confidence intervals calculated in R using $1 \times 10^5$ bootstrap replications.

For FRAP analysis, an ImageJ macro was used to define and measure the GFP intensity (mean pixel density) in the FRAP region, background, and whole cell. These data and time stamps from the open microscopy environment were fed into IgorPro for processing. The background-subtracted intensities for the FRAP region and whole cell were used to calculate a ratio (to correct for bleaching of molecules induced by the procedure). These values were paired with the time stamps and scaled so that the intensity after bleach was 0 and an average of the first five images was 1, and then an interpolated average was created. Fits to individual traces were also calculated using a script. Double exponential function was used for fitting, since this gave better fits than a single exponential, particularly for GFP-TPD54, and so that all conditions were fitted in the same way for comparison.

All figures were made with either Fiji or Igor Pro 8 (WaveMetrics) and assembled using Adobe Illustrator.

### Data and software availability

The data for proteomics, volcano plot, FRAP data, and EM segmentation coordinates are available, together with code and scripts for analysis, at https://doi.org/10.5281/zenodo.3366083 (Royle, 2019).

### Online supplemental material

Fig. S1 shows the characterization of the GFP-TPD54 knock-in cell line. Fig. S2 demonstrates the localization of mCherry-TPD54 in HeLa cells. Fig. S3 shows the characterization of TPD54 knockout cell lines. Fig. S4 shows FRAP analysis of GFP and GFP-TPD54. Fig. S5 shows the co-rerouting of Rab GTPases with TPD54, the co-rerouting of endogenous Rab1a, and the co-rerouting of TPD54 with Rab GTPases. Videos 1 and 2 show SBP-EGFP–E-cadherin RUSH imaging in control or TPD54-depleted cells, respectively. Video 3 demonstrates rerouting of mCherry-FKBP-TPD54 to mitochondria. Video 4 shows imaging of subresolution vesicle fluorescence. Video 5 demonstrates FRAP of GFP or GFP-TPD54. Video 6 shows co-rerouting of a GFP-Rab with mCherry-FKBP-TPD54.

## Acknowledgments

We thank Patrick Caswell, Francis Barr, and Andrew Peden for valuable discussion and reagents. We thank members of the laboratory for constructive criticism, and especially Cecilia Velasco Dominguez, Oliver Sinfield, and Andrew Fielding for their earlier work on TPD52 and TPD54. The authors thank Erick Martins Ratamero and Claire Mitchell of the Computing and Advanced Microscopy Unit for their support and assistance in this work.

G. Larocque was supported by Fonds de Recherche du Québec – Nature et Technologies and University of Warwick Chancellor's Award. N.J. Carter was supported by a Wellcome Trust Investigator Award to Robert Cross. The work was supported by the UK Medical Research Council grant MR/P018947/1.

The authors declare no competing financial interests.

Author contributions: G. Larocque did all experimental work, analyzed data, and wrote the paper. P.J. La-Borde made the GFP-TPD54 cell line. N.I. Clarke performed correlative-light EM imaging. N.J. Carter performed STORM imaging with G. Larocque. S.J. Royle analyzed data, wrote computer code, and wrote the paper.

Submitted: 7 December 2018

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

# Supplemental material

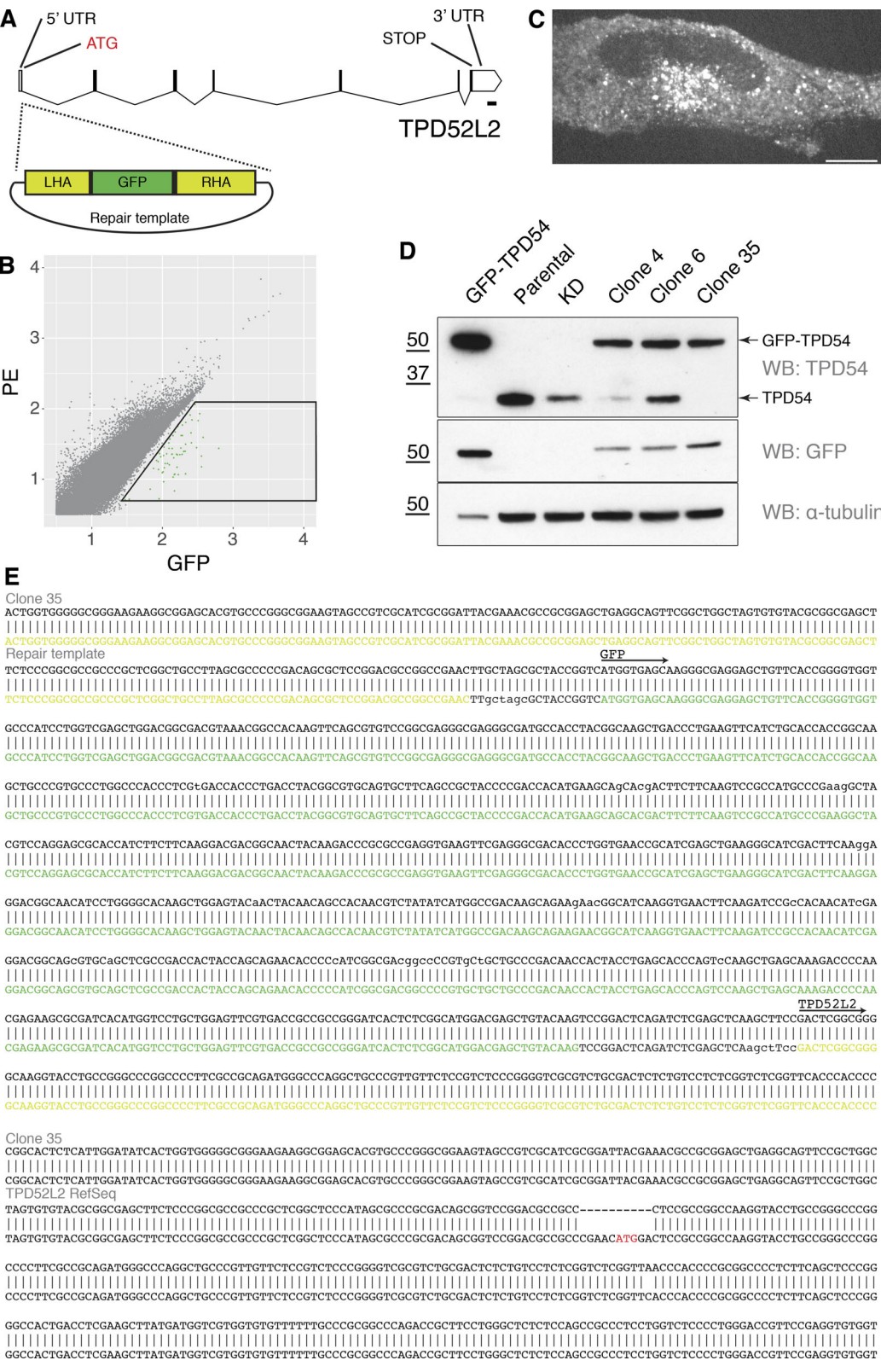

Figure S1. **Characterization of the GFP-TPD54 knock-in cell line. (A)** Gene map of TPD52L2 and location of GFP tagging. **(B)** FACS plot. GFP-positive cells in the indicated gate were recovered and characterized. **(C)** Representative confocal micrograph of an example GFP-TPD54 knock-in clone. Scale bar, 10 µm. **(D)** Clones were validated by Western blot (WB). Cells overexpressing GFP-TPD54, parental HeLa cells, TPD54-depleted cells, and three different clones are shown. Clone 35 exhibited the desired band profile. A single GFP-TPD54 band detected by blotting for TPD54 and GFP, with no untagged TPD54. Tubulin is shown as a loading control; note that one tenth of the GFP-TPD54 transfected sample was loaded. **(E)** Sequencing the TPD52L2 locus in clone 35. Two bands were amplified using primers flanking the integration site. The first sequence shows integration of monomeric GFP between the homology arms, giving GFP-TPD54. The second sequence shows that clone 35 is null at the other allele. KD, knockdown; PE, phycoerythrin.

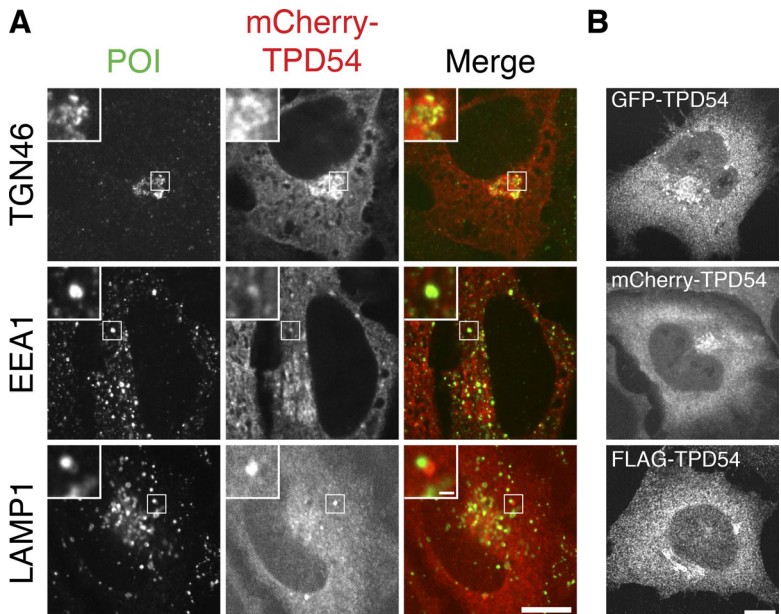

Figure S2. **Overexpressed TPD54 colocalizes with membrane trafficking components. (A)** Representative confocal micrographs comparing the subcellular distributions of mCherry-TPD54 with those of the Golgi apparatus marker TGN46, the early endosomal marker EEA1, or the lysosomal marker LAMP1. **(B)** Examples of the similar subcellular distribution of TPD54 in cells expressing GFP-TPD54, mCherry-TPD54, or FLAG-TPD54 (detected by immunofluorescence). Insets, 3× zoom. Scale bars, 10 µm, 1 µm (insets). POI, proteins of interest.

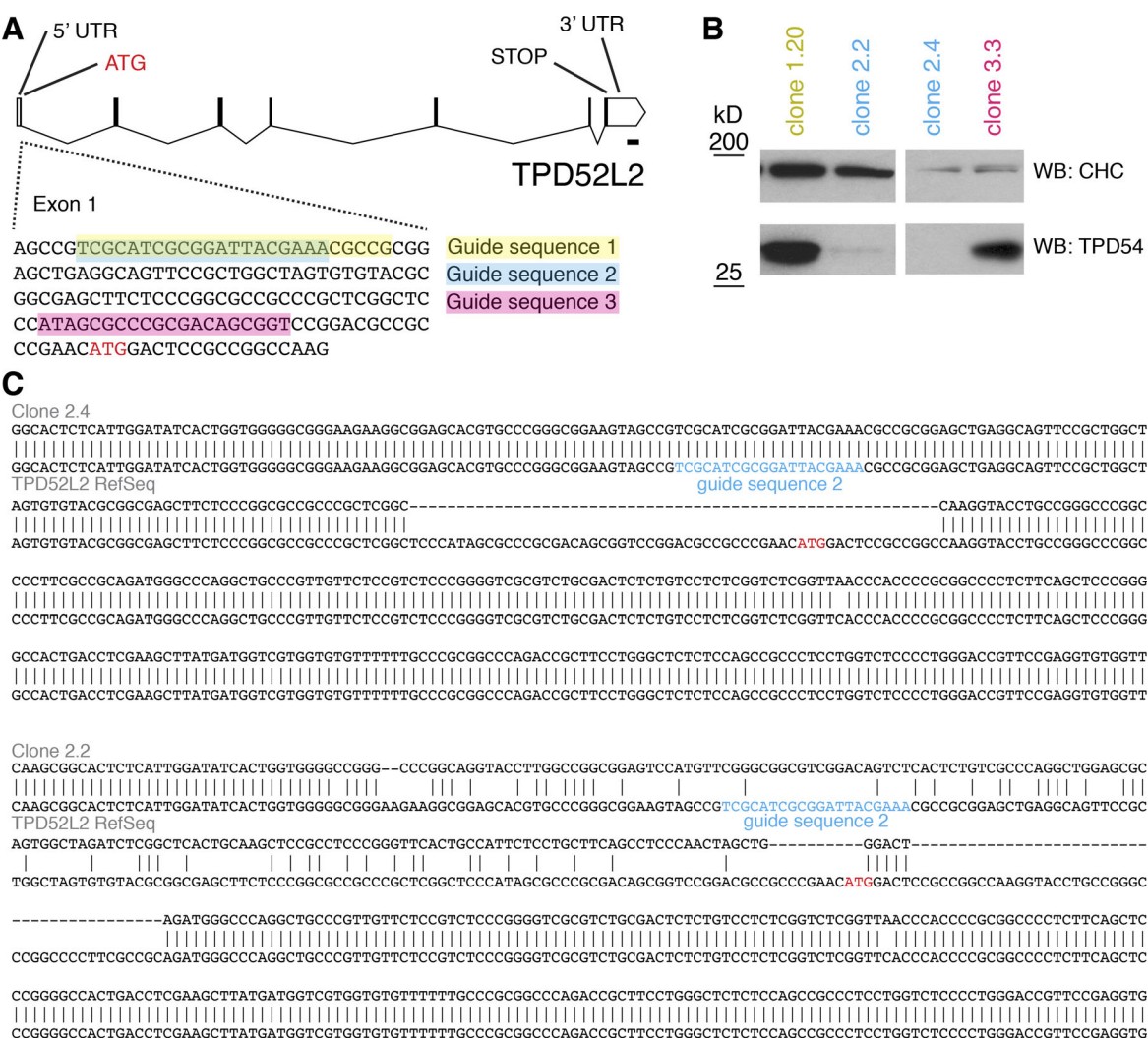

Figure S3.   **Targeted disruption of TPD54 gene in HeLa cells using CRISPR/Cas9. (A)** Three guides were designed to target the TPD52L2 locus. **(B)** Single-cell clones were isolated and screened by Western blotting. Two clones, 2.2 and 2.4, showed loss of TPD54 expression. **(C)** Sequencing of PCR amplicons using primers flanking the CRISPR/Cas9 targeting site revealed disruption of the locus in clones 2.2 and 2.4. Sequencing of the top three most similar protospacer adjacent motif (PAM) sequences in the genome showed no change from the parental sequence. CHC, clathrin heavy chain.

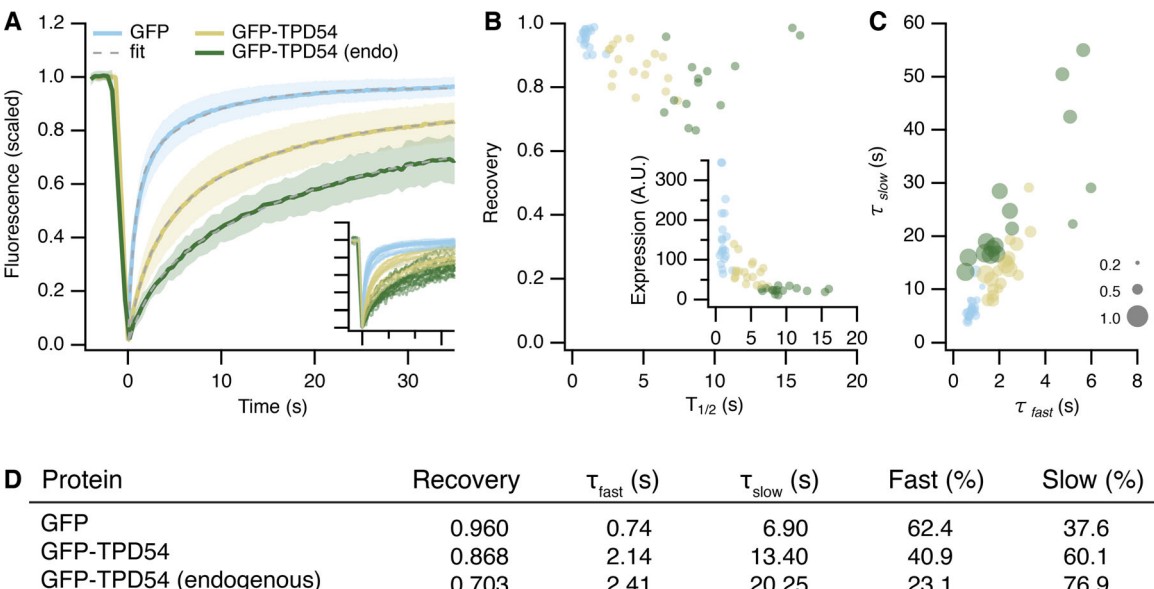

| Protein | Recovery | $\tau_{fast}$ (s) | $\tau_{slow}$ (s) | Fast (%) | Slow (%) |
|---|---|---|---|---|---|
| GFP | 0.960 | 0.74 | 6.90 | 62.4 | 37.6 |
| GFP-TPD54 | 0.868 | 2.14 | 13.40 | 40.9 | 60.1 |
| GFP-TPD54 (endogenous) | 0.703 | 2.41 | 20.25 | 23.1 | 76.9 |

Figure S4. **FRAP analysis of GFP-TPD54. (A)** FRAP data for GFP-TPD54 in knock-in cells (endo), expressed GFP, or GFP-TPD54 in parental cells. Lines and shaded areas show mean ± 1 SD. Dashed line shows a double-exponential function fitted to the average data. The fit coefficients are summarized in D. Inset: FRAP data from different cells colored as indicated and displayed on the same axes range. **(B)** Recovery (mobile fraction) of individual fits to FRAP data plotted against $t_{1/2}$. Inset: initial cellular fluorescence as a function of $t_{1/2}$. **(C)** Plot to show FRAP kinetics in individual cells. $\tau_{slow}$ is plotted against $\tau_{fast}$, and marker size indicates the fraction recovered by the slow process. Markers represent individual cells, and colors indicate the protein being imaged. **(D)** Kinetics of FRAP. FRAP kinetics were much slower for GFP-TPD54 (either expressed or endogenous) compared with GFP, suggesting GFP-TPD54 is bound to membranes. There were two phases of GFP-TPD54 recovery: a small, fast process ($\tau$ = 2 s) with the majority of recovery via a slow process, which was in the order of tens of seconds. These kinetics were consistent with the majority of TPD54 binding to subcellular structures, with a minor fraction being cytosolic. Analysis of individual FRAP traces showed that in cells expressing higher levels of GFP-TPD54, FRAP was faster and that this was due to a larger fraction recovering via the fast process. This is consistent with overexpression saturating the membrane-bound population and causing some TPD54 to be cytosolic. Note that the kinetics of TPD54 rerouting were best described as a single process ($\tau$ = ~40 s), presumably corresponding to vesicle capture, with no detectable faster component that would suggest a diffusible pool of TPD54 in the cytosol.

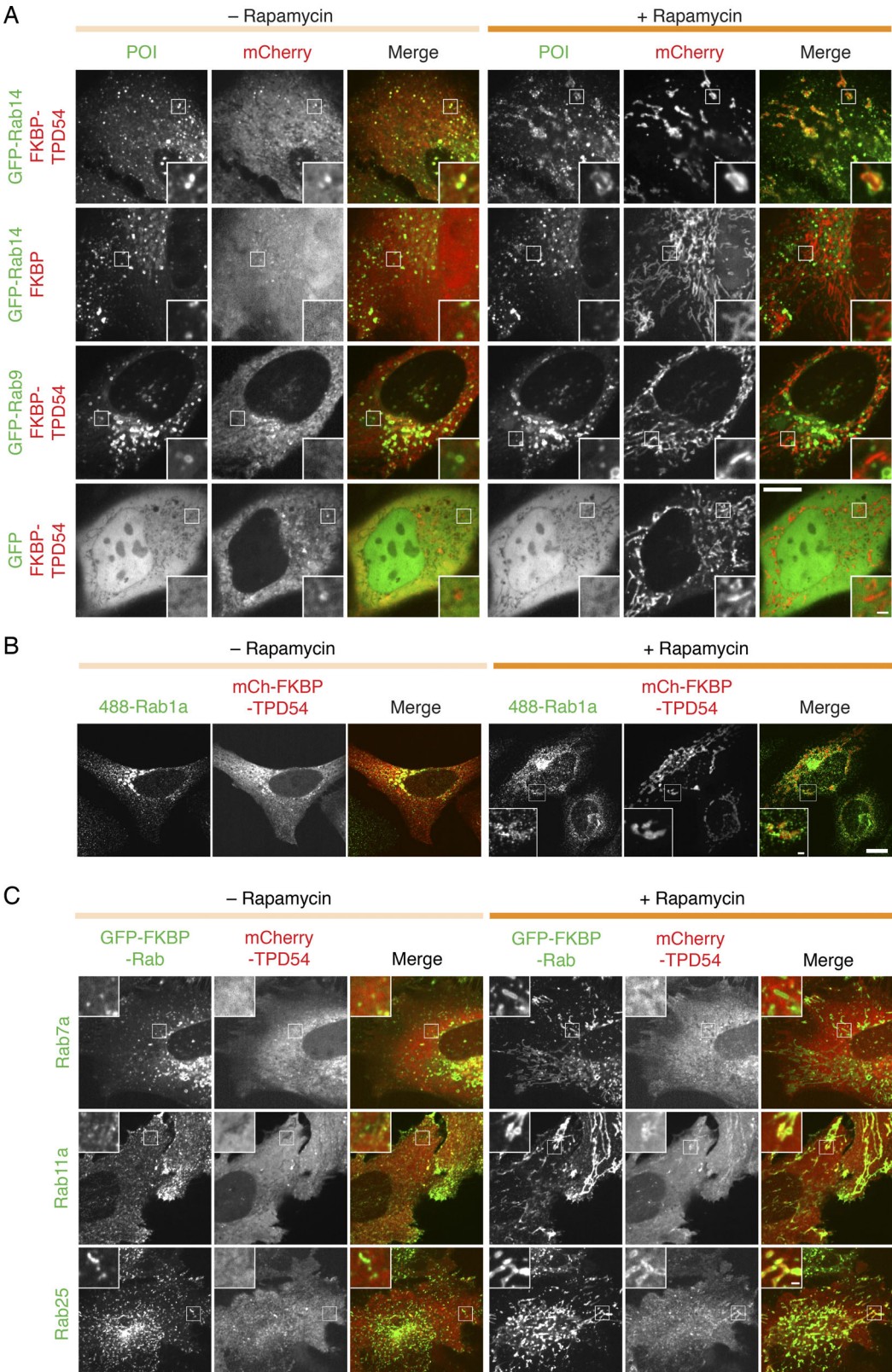

Figure S5. **Co-rerouting of Rab GTPases and TPD54 to mitochondria. (A)** Representative micrographs showing the co-rerouting of GFP-Rab14, but not GFP or GFP-Rab9a, after rerouting of mCherry-FKBP-TPD54 to dark Mitotrap by addition of 200 nM rapamycin. Note that GFP-Rab14 localization is unaffected by rerouting of mCherry-FKBP. **(B)** Representative micrographs showing the co-rerouting of Rab1a detected by immunofluorescence after rerouting of mCherry-FKBP-TPD54 to dark Mitotrap by addition of 200 nM rapamycin. **(C)** Two positive hits (Rab11a and Rab25) and a negative Rab (Rab7a) were tested for TPD54 co-rerouting. Micrographs of cells before and after rerouting the indicated GFP-FKBP-Rab to dark MitoTrap in cells also expressing mCherry-TPD54. Insets, 3× zoom. Scale bars, 10 μm, 1 μm (insets). POI, proteins of interest.

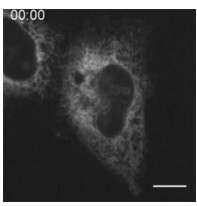

Video 1.   **SBP-EGFP–E-cadherin RUSH imaging in control cells.** Live-cell confocal microscopy of RUSH assay; biotin is added at time 0. Still images from this video are shown in Fig. 2. Time, hours and minutes. Scale bar, 10 μm.

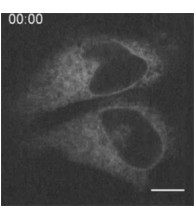

Video 2.   **SBP-EGFP–E-cadherin RUSH imaging in TPD54-depleted cells.** Live-cell confocal microscopy of RUSH assay; biotin is added at time 0. Still images from this video are shown in Fig. 2. Time, hours and minutes. Scale bar, 10 μm.

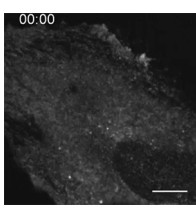

Video 3.   **Rerouting mCherry-FKBP-TPD54 to mitochondria in cells coexpressing MitoTrap.** Live-cell confocal microscopy of rerouting assay; mCherry-FKBP-TPD54 is rerouted to MitoTrap using rapamycin 200 nM, applied at 10 s. Still images from this video are shown in Fig. 4. Time, minutes and seconds. Scale bar, 10 μm.

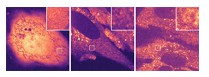

Video 4.   **Imaging subresolution vesicle fluorescence.** Live-cell confocal microscopy of cells expressing GFP (left) or GFP-TPD54 (center); GFP-TPD54 knock-in cells are shown to the right. Captured at 120 ms per frame. Playback, 10 frames per second. ROI is 3.5 μm × 3.5 μm, expansion is fivefold.

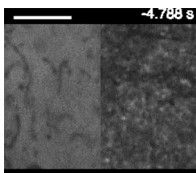

Video 5.   **FRAP.** FRAP of GFP (left) or GFP-TPD54 (right) expressed in HeLa cells. Bleach area is a rectangle inset by 0.8 μm. Time is indicated. Scale bar, 5 μm.

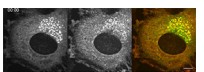

Video 6.   **Co-rerouting of a GFP-Rab with mCherry-FKBP-TPD54.** Live-cell confocal microscopy of rerouting assay. Co-rerouting of GFP-Rab30 (left; shown in right panel in green) with mCherry-FKBP-TPD54 (middle; shown in right panel in red) to dark MitoTrap (not shown) using rapamycin 200 nM, applied at 10 s. Time, minutes and seconds. Scale bar, 10 μm.

