## [Reviewer comments · The Journal of Cell Biology]

Tumor Protein D54 defines a new class of intracellular transport vesicles

Gabrielle Larocque, Penelope La-Borde, Nicholas Clarke, Nicholas Carter, and Stephen Royle

Corresponding Author(s): Stephen Royle, University of Warwick

Review Timeline:

Submission Date:	2018-12-07
Editorial Decision:	2019-01-13
Revision Received:	2019-09-02
Editorial Decision:	2019-09-30
Revision Received:	2019-10-02

Monitoring Editor: Sean Munro

Scientific Editor: Melina Casadio

Transaction Report:

DOI: <https://doi.org/N/A>

January 13, 2019

Re: JCB manuscript #201812044

Dr. Stephen J Royle
University of Warwick
Centre for Mechanochemical Cell Biology
Division of Biomedical Cell Biology Warwick Medical School
Gibbet Hill Road
Coventry CV4 7AL
United Kingdom

Dear Steve,

Thank you for submitting your manuscript entitled "Tumor Protein D54 is a promiscuous Rab effector". The manuscript has been evaluated by expert reviewers, whose reports are appended below. We sincerely apologize for the delay in communicating our decision to you. Unfortunately, after an assessment of the reviewer feedback, our editorial decision is against publication in JCB.

As you can see, the referees acknowledge the potential interest of the findings but they also raise a number of concerns. This reflects that your observations, although striking, involve phenomena that are unexpected. These concerns include the apparent ability of TPD54 to bind to so many different Rabs, and yet the interaction has eluded detection in previous studies of these Rabs; the accumulation of vesicles that are smaller than any known population of intracellular transport vesicles; and the uncertainty as to the mechanism by which Rabs accumulate on TPD54-coated mitochondria.

The extensive nature of these concerns means that the paper is not suitable for the JCB in its current form. We feel that the points raised by the reviewers are more substantial than can be addressed in a typical revision period. If you wish to expedite publication of the current data, it may be best to pursue publication at another journal.

However, given interest in the topic, we would be open to resubmission to JCB of a significantly revised and extended manuscript that fully addresses the reviewers' concerns and is subject to further peer-review. If you would like to resubmit this work to JCB, please contact the journal office to discuss an appeal of this decision or you may submit an appeal directly through our manuscript submission system. Please note that priority and novelty would be reassessed at resubmission.

Indeed, if you could obtain substantial new data to address the points raised by the reviewers, we would be willing to consider a future resubmission on TPD54. Looking at the reviewers' comments, it seems most critical to solidify the evidence for the key observations. In particular, the evidence for direct binding to Rabs could be strengthened, and the nature of the vesicles could be investigated in more depth. For Rab binding, testing more Rabs seems like a good suggestion, especially as the in vitro binding assay shows a rather small difference between the GDP and GTP versions of Rab1. I also noticed that your original pull-down that identified interactions with Rabs was performed in the presence of EDTA, which will deplete magnesium and hence greatly reduce the levels of GTP on the Rabs. Perhaps this could be repeated in conditions compatible with GTP-binding. The reviewers

are also concerned as to whether accumulation of the Rabs on TPD54-coated mitochondria reflects direct binding, or perhaps the Rab simply being on the captured vesicle. Here you could try forms of the Rab that lack the lipid anchor and so should not be membrane-bound. Likewise, you could probe with antibodies against endogenous Rabs as these should be titrated away from their normal site of action. The other area where more evidence would help is the nature of the vesicles. This is harder, but it seems important to look at more endogenous cargo proteins and other markers, as the reviewers suggest. Of course, the other big question is the actual function of TPD54. Addressing this is probably unrealistic in an initial manuscript, but this requires that the new phenomena in the paper are as robust and unambiguous as possible.

Thus, if you can characterize more extensively TPD54's interactions with Rabs and reveal more about the cellular mechanisms that are reflected in the accumulation of vesicles around TPD54-coated mitochondria, we feel that a paper would have a better chance of approval from reviewers at the JCB.

Regardless of how you choose to proceed, we hope that the comments below will prove constructive as your work progresses. We would be happy to discuss the reviewer comments further once you've had a chance to consider the points raised in this letter. You can contact the journal office with any questions, cellbio@rockefeller.edu or call (212) 327-8588.

Thank you for thinking of JCB as an appropriate place to publish your work.

Sincerely,

Sean
Sean Munro, PhD
Monitoring Editor, Journal of Cell Biology

Melina Casadio, PhD
Senior Scientific Editor, Journal of Cell Biology

Reviewer #1 (Comments to the Authors (Required)):

This is an interesting contribution providing evidence that Tumor Protein D54 has effector-typical properties for a number of Rab proteins. Tumor Protein D54 is a member of the Tumor Protein D52 (TPD52) family, most of which are expressed ubiquitously, and at least in the case of TPD52 are highly expressed in several cancers. Their role remains fairly obscure, although they have been implicated in membrane trafficking. TPD54, the subject of this article, is particularly highly expressed.

Localisation studies of TPD54 reported here show association with the Golgi apparatus and endosomes, and diffuse distribution throughout the cytoplasm. Using Golgi and endosomal markers, the authors conclude that there is colocalization with TPD54, although inspection of Fig. 1 A suggests that this is better described as a localization to the same organelle rather than a colocalization, particularly for the Golgi marker (GalT). The authors should comment on this. Subsequent pulldown experiments provided evidence that TPD54 interacts with members of the TPD52 family (as expected, since they are known to dimerize and heterodimerize), but also with Rab2a, Rab5c and Rab14. This suggested a possible involvement with anterograde and recycling pathways, which was confirmed in a general using TPD54-depleted cells. It was already known

that TPD52 interacts with Rab5c (quoted in the manuscript), and there was earlier indirect evidence of interaction of TPD52 with Rab5, Rab6 and Rab9 (Zhang, Ding and Barbieri, JBC (20016)) which should be cited in the paper.

Mislocation of TPD54 to mitochondria resulted in capture of small vesicles, affected mitochondria morphology and caused aggregation. Essentially all labelled TPD54 was rerouted to mitochondria, with the "cytosolic" fraction being recruited more rapidly than that on endosomes. Using a library of 43 labelled Rabs, the authors were then able to show that 16 Rabs relocated to mitochondria with TPD54, suggesting an interaction (direct or indirect). In a reciprocal relocation experiment in which 3 Rabs (Rab11a, Rab 25 and Rab7a) were recruited to mitochondria, TPD54 was also mislocated in the case of Rab11a and Rab25, which were both positive in the "forward" screen, but not in the case of Rab7, which was negative in the first screen, confirming the general interpretation of interaction between the 16 Rabs (or at least the vesicles on which they are located) and TPD54. Analysis of the group of Rabs identified suggested that they are all involved in anterograde or recycling pathways, although not all Rabs known to be involved on these pathways were identified as TPD54 interactors.

Direct evidence for interaction of Rab1a in its active form (using GppNHp) with TPD54 was obtained using recombinant proteins, whereas the inactive (GDP) form did not interact. Rab1a was positive in the rerouting experiment. Rab6a, which was negative in the screen did not interact. Thus, at least in the case of Rab1a, TPD54 has the properties of an effector. Ideally, it would have been good to have a few more positive and negative examples to give the conclusions a solid basis. One more result helps in this respect: using "active" and "inactive" mutants of Rab 30, Rab1a and Rab6a, only the active (or wild type) variants of Rab30 and Rab1a could be rerouted to mitochondria, while the inactive mutants and none of the Rab6 variants relocated.

Finally, evidence is given that TPD54 is active in promoting vesicle fusion, since knock-down or knock-out of TPD54 led to dispersion of the trans-Golgi network. This interpretation would need considerable further evidence to allow a definite conclusion with mechanistic insights on this question.

In general, the main conclusion of the paper (that TPD54 is a promiscuous Rab effector) appears to be well founded, although more biochemical evidence would make the argument more compelling. One point that perhaps needs some clarifying arguments is that made in the second sentence of the discussion. Did the authors really show that TPD54 is present on numerous (types of) vesicles? Is this not an interpretation of the evidence that at least 16 Rabs are relocated with TPD54 to mitochondria? Do we know how much of this is due to vesicle recruitment, and how much might come from recruitment of a genuine cytosolic Rab fraction (presumably GDI bound)? Perhaps there are arguments that I have missed on these points. This is related to the commonly arising question of "what recruits what" in terms of Rab composition of membranes and the general principle of membrane identity.

Reviewer #2 (Comments to the Authors (Required)):

The authors study the abundant but mechanistically uncharacterized protein TPD54. Their principal conclusions are as follows. (1) TPD54 associates with vesicles. (2) These vesicles are not obviously coated and are very small, with a diameter of about 30 nm. (3) These vesicles contain some cargoes (e.g., with dileucine sorting motifs) but not others. (4) These vesicles contain some SNAREs (e.g., VAMPs 2, 3, 7, and 8) but not others (VAMP4, five tested Q-SNAREs). (5) TPD54 interacts with at least 16 Rabs. (6) In the case of Rab1a, this interaction is direct. (7) In the cases of Rab30 and Rab1a, this interaction is specific to the active, GTP-bound form of the Rab. (8) Depleting TPD54 leads to anterograde and recycling trafficking defects and Golgi dispersal.

There are a number of surprises here. First, the vesicles. They are smaller than well-characterized vesicular carriers (e.g., COPI, COPII, clathrin-coated vesicles) and the authors boldly propose that they are an overlooked class of vesicles that have heretofore evaded study. Second, the large number of VAMPs that appear to co-populate TPD54 vesicles. This, along with the functional consequences of depleting TPD54, suggests that TPD54 vesicles are involved in trafficking to various destinations, making it all the more remarkable that they have not previously been reported. Third, the promiscuous Rab interactome. Given the modest size of TPD54, it's rather incredible that it is a potential effector for over a dozen Rabs. This suggests a key conserved function. What is it, and how has it escaped attention until now? All of which is to say that there is a much of potential interest to the reader of JCB in this manuscript.

Major points

1. The main evidence for Rab-TPD54 interactions comes from the observation that rerouting TPD54 to the mitochondria causes significant rerouting of 16 different Rabs. The cartoon in Fig. 3D suggests that the interaction between TPD54 and the rerouted Rab is direct, but it could also be indirect - any protein associated with the rerouted vesicle would presumably be rerouted along with it. The only evidence in favor of a direct interaction is the pull-down assay (Fig. 9a), and it only contains one positive result (for GppNHp-loaded GST-Rab1a). Given the major focus of this paper on TPD54 as a Rab effector, and the mechanistic consequences thereof, it would be very reassuring to have further evidence for direct interactions.

2. The striking (and central to this manuscript) idea that swarms of previously undetected 30-nm vesicles are conveying cargo hither and yon within the cell would greatly benefit from further direct experimental support. An 'obvious' idea would be immunogold labeling, in conjunction with rerouting experiments.

Minor points

1. As GFP-tagging can significantly influence the intracellular localization of proteins, it would be reassuring to see that smaller tags (e.g. HA, flag) lead to the same localization.

2. The transferrin uptake/recycling assay should be presented more clearly; for example, which parts of Fig. 2h correspond to what phase of the assay?

3. As the authors suggest that (most of the) TPD54 is actually not cytoplasmic but instead bound to small vesicles, does the initial rerouting of cytoplasmic mCherry-FKBP-TPD54 (and the deduced kinetics thereof) represent an overexpression artifact?

4. I didn't entirely understand the 'flickering' argument for TPD54 being on vesicles, and in any case it seems that the cited video (SV4) actually represents the FRAP experiment discussed subsequently.

Reviewer #3 (Comments to the Authors (Required)):

Larocque et al investigate the role of tumour protein D54, member of the TPD52-like protein family, concerning its function in anterograde membrane trafficking and endosomal recycling. Using a wide

range of cell biological tools, such as endogenous protein tagging, CRISPR-mediated KO cell lines, co-immunoprecipitation and rapamycin inducible mis-localization, they show that 1) several distinct Rab GTPases co-immunoprecipitated with TPD54 and 2) mis-localization of TPD54 to mitochondria lead to an accumulation of small vesicles on mitochondria concomitant with rerouting of different Rab GTPases to mitochondria. Combining the cell biological approach with biochemistry and EM the authors hypothesize that TPD54 associates with an unknown class of small vesicles and directly interacts with different Rab GTPases, constituting a promiscuous Rab effector.

While the key message of the study is provocative, the manuscript gives the impression of pieces of data put together in a rather careless and sloppy fashion. There are flaws and weaknesses at different levels, starting from the overall composition, which lacks coherence. On several occasions the work is inconsistent and it is not well written. Besides the presentation, I have serious concerns about the scientific quality of the data. The biochemistry on which the whole story is built is weak and incomplete. Furthermore, there is lack of reasoning, explanation and experimental care in different parts of the manuscript. For example, important controls are missing. The interpretation of the results seems to be biased towards the central hypothesis on several occasions, neglecting the possibility that serious perturbations such as overexpression of proteins and RNAi-mediated knock-down may also lead to the observation of off-target effects and overexpression artefacts. The extent of work and the number experiments needed for this manuscript to form a coherent, comprehensive and scientifically solid paper is far beyond a normal revision, I therefore recommend rejection of this manuscript.

Major comments

1. The title of the manuscript "Tumor Protein D54 is a promiscuous Rab effector" comprises a strong statement about the direct and nucleotide dependent interaction of TPD54 with Rab GTPases which the authors do not convincingly show anywhere in the manuscript. Biochemical indication is only provided for the interaction of TPD54 with Rab1a using a GST pulldown assay. For other Rabs the authors only show co-rerouting experiments, which are not sufficient to prove a direct and nucleotide dependent interaction and hence are not suitable to identify Rab effectors. Even though the requirements for Rab effectors are clearly mentioned in the manuscript, the authors base their statement on a single interaction assay with Rab1a and co-rerouting experiments. To prove this hypothesis, interaction assays for most if not the full set of Rab effectors tested in the co-rerouting experiments combined with a thorough biochemical characterization of the components (see point 2) would be necessary.

2. The interaction assays provided in the manuscript seem to be weak, erroneous in experimental details and not properly described in the methods section.

Firstly, why do the authors use an MBP-TPD54-His variant of the protein to perform the interaction assay, even though the MBP tag is not necessary for the experiment? Details about the construct design are omitted in the manuscript and should be clarified. As the MBP tag is often used to keep proteins soluble, a thorough biochemical characterization is required to ensure protein quality.

Secondly, the authors state that "proteins were eluted from the beads in Laemmli buffer...". This experimental approach potentially leads to a misinterpretation of the results, as Laemmli buffer also elutes proteins that are aggregated on the beads, instead of interacting with the GST-Rab. This can be avoided by eluting with a Glutathione containing elution buffer (20 mM GSH) and this needs to be shown to strengthen the authors' point.

Next problem, after purification of GST, GST-Rab1a, GST-Rab6a and MBP-TPD54-His a buffer exchange is required to perform the described experiments (GST interaction assays). However, this step is not described in the methods section nor mentioned in the manuscript. Was it done by

dialysis, size exclusion chromatography or a different method?

How was the successful loading of the Rab GTPases checked? The authors do not provide any detail of how quantitative loading was ensured.

All these points are especially important as the main claim of the paper is the direct interaction of TPD54 with different Rab GTPases as an effector, i.e. in a nucleotide-dependent manner.

3. Was the GFP-TPD54 cell line described in the first section of the results tested using the transferrin uptake assay mentioned in the second paragraph of the results? If not please check if the GFP-TPD54 cell behaves similarly as HeLa wt.

4. How many different siRNAs against TPD54 were used in the knock-down experiments (Figure 2)? How many of these gave a consistent phenotype? The authors need to provide evidence that the phenotype is not due to off-target effects of the siRNA.

5. The authors argue that a flickering in the cytoplasmic TPD54 signal is observed in contrast to GFP. After a quantification of diffusion rates using FRAP it is stated that "These experiments indicate that the freely diffusing pool of TPD54 is minimal and the majority of TPD54 is associated with small vesicles below the resolution limit." Would the authors please explain why TPD54 needs to be bound to small vesicles and what would be the result of the FRAP experiment if TPD54 were to unspecifically interact with any membrane? Can these small vesicles be visualized by EM under normal conditions? The overexpression of mCherry-FKBP-TPD54 may induce the formation of these vesicles which are subsequently targeted to mitochondria by rapamycin.

6. Following the observation of small vesicles on mitochondria, the authors assess their functionality using co-rerouting of overexpressed model cargo and co-localization of SNARE proteins. How do these experiments show that the vesicles are functional? What is the function of these vesicles? The authors indicate in the discussion that these vesicles are a so far un-described and unidentified. If true, it is necessary to identify these vesicles in other cell lines under normal conditions and to further characterize them, otherwise this idea stands in the air without any significance.

What is the situation in non-cancer cell lines where TPD54 is not overexpressed, as observed in HeLa cells? The experiments are not very well explained and it seems to me that the experimental design is not suitable to answer the questions posed in the manuscript.

7. How does overexpression of Rab GTPases affect the rerouting to mitochondria upon addition of rapamycin? Is the rerouting still observed in cell lines expressing near endogenous levels of fluorescent Rab GTPases?

8. "Second, we marked up our hits on a phylogenetic tree of the Rab collection (Supplementary Figure S4B). This diagram showed that the positive hits generally belonged to a clade, with a common ancestral sequence. This suggested to us that TPD54 binds directly to these particular Rab GTPases rather than being independently localized on the same vesicle." Can the authors please explain how a common ancestral sequence points towards a direct interaction between TPD54 and a set of Rab GTPases? Do the authors implicate the common ancestral sequence to be the interface? Has this been verified by mutational studies?

9. In the last section of the results it is not clear how a dispersal phenotype of the Golgi indicates a role of TPD54 in vesicle fusion. The authors need to explain their line of argumentation and provide further evidence for their hypothesis.

Minor:

1. The manuscript is written in a rather sloppy and partly unscientific fashion. The introduction reads very general as small chapters summarizing textbook knowledge, rather than a dedicated introduction to this specific paper. Some examples:

"Despite its abundance, there are virtually no published data on the cell biology of TPD54." What does that mean? If there is no data published it can be stated like that, if there is data please cite.

"To our frustration, the Golgi dispersal phenotype in both knockout clones disappeared with repeated passaging, which might be explained by compensation for the chronic loss of TPD54 in knockout cells. These experiments confirm that the dispersal phenotype is due to loss of TPD54 specifically and is not the result of off-target action. Furthermore, they indicate that the effector function for TPD54 is likely to be in the promotion of vesicle fusion."

What does it mean if the phenotype is lost? Can the other TPD52-like proteins compensate the effect?

2. A figure with a schematic alignment of the different TPD52-like proteins would be helpful

3. Why do the authors investigate the interaction of TPD54 with Rab1 by GST pulldown assays, when Rab14, Rab2a and Rab5c were identified as significant hits by co-immunoprecipitation?

Reviewer #0:

A brief summary of the changes made, in order of importance:

- The title, abstract and discussion have changed. Focus of the paper is different
- Removal of Rab binding data and testing QL and S/TN mutants by rerouting
- Addition of STORM imaging of INVs (Figure 6)
- Addition of quantification of spatiotemporal variance of GFP-TPD54 fluorescence (Figure 6)
- Addition of two other siRNAs to RUSH figure (Figure 2)
- Removal of confusing schematic figures
- Focus on rerouting as INV recruitment to mitochondria and co-rerouting as the movement of INV passengers
- Supplementary figure to show co-rerouting of endogenous Rab1a (Figure S6)
- Supplementary figure to show FLAG-TPD54 is similar to GFP/mCherry (Figure S2)
- Model figure as a final figure (Figure 10)
- FRAP moved to Supplementary Info (Figure S4, Table S1)
- All phenotypic analysis grouped together (Figures 2-3)

Indeed, if you could obtain substantial new data to address the points raised by the reviewers, we would be willing to consider a future resubmission on TPD54. Looking at the reviewers' comments, it seems most critical to solidify the evidence for the key observations. In particular, the evidence for direct binding to Rabs could be strengthened, and the nature of the vesicles could be investigated in more depth. For Rab binding, testing more Rabs seems like a good suggestion, especially as the *in vitro* binding assay shows a rather small difference between the GDP and GTP versions of Rab1. I also noticed that your original pull-down that identified interactions with Rabs was performed in the presence of EDTA, which will deplete magnesium and hence greatly reduce the levels of GTP on the Rabs. Perhaps this could be repeated in conditions compatible with GTP-binding.

In addressing this point, which was also raised by all Reviewers, we realized that our conclusion that TPD54 was a promiscuous Rab effector was premature. We have changed the focus of the paper as a result. A brief summary of the work we did:

- We cloned a further 13 GST-tagged Rabs for this revision work (Rab7a, Rab9a, Rab10, Rab11a, Rab12, Rab14, Rab19, Rab21, Rab25, Rab26, Rab30, Rab33b and Rab43).
- TPD54 binding experiments were done with Rab7a, Rab11a, Rab12, Rab21, Rab26, Rab30 and Rab33b. Expression of the remaining Rabs was insufficient for binding assays (Rab9a, Rab10, Rab14, Rab19, Rab25 and Rab43).
- Again, we tested for "effector binding". Does TPD54 bind preferentially to the active (GppNHp-bound) form of the Rab compared to the inactive (GDP-bound)?
- The results were more variable than what we observed with Rab6a and Rab1a, shown in the previous manuscript.
- TPD54 bound to Rab11a, Rab12 and Rab33b, but showed no preference between the active and inactive forms (n=2-3). These Rabs were positive in our screen.
- More TPD54 bound to the inactive forms of Rab26 and Rab30 than the active forms (n=2). These Rabs were positive in our screen.
- TPD54 was seen to bind to Rab7a and Rab21 (n=1). These Rabs were negative in our screen. Here there was more binding of the inactive forms of Rab7a and Rab21 compared to the active forms.

Our conclusion is that we got "lucky" with our experiments on Rab1a/Rab6a and this misled us into thinking that positive hits in our Rab screen were due to TPD54 being an effector at all of the co-rerouted Rabs. Now that we have tested a wider variety of Rabs in the same way, we can see that this extrapolation is not valid.

In the revised paper we have removed the TPD54/Rab binding experiments and we have changed the focus to be on the vesicles themselves rather than on TPD54. In the discussion we leave open the possibility that TPD54 binds Rabs. After all, we are still showing the mass spec data that TPD54 binds Rab2a, Rab5c, Rab14. However, the conclusion about the Rab screen is parsimonious: positive Rabs are passengers on the rerouted INVs with no conclusion possible about direct binding.

The reviewers are also concerned as to whether accumulation of the Rabs on TPD54-coated mitochondria reflects direct binding, or perhaps the Rab simply being on the captured vesicle. Here you could try forms of the Rab that lack the lipid anchor and so should not be membrane-bound.

As described above we now state that any proteins that co-reroute with TPD54 must be considered passengers on the INV and not necessarily interactors. In this context, Rabs are no different from the SNAREs or the cargo that we also co-reroute. Nonetheless we did the experiment that you suggested (Reviewer Figure R1). These results agree with the idea that the Rab is a passenger rather than a TPD54-binding protein. However, we have not included this data in the paper because there are other ways that this experiment may be interpreted. Without membrane attachment, the Rab will not gain GTP, even as a QL mutant (Barr, *J Cell Biol* 2013). Therefore this experiment doesn't rule out the possibility TPD54 can bind to active Rabs that are not on vesicles.

Likewise, you could probe with antibodies against endogenous Rabs as these should be titrated away from their normal site of action. The other area where more evidence would help is the nature of the vesicles. This is harder, but it seems important to look at more endogenous cargo proteins and other markers, as the reviewers suggest.

We have added a new figure (Supplementary Figure S6) that shows the extent of rerouting of endogenous Rab1a. We also show that endogenous cargo (CIMPR) and endogenous SNAREs are co-rerouted (Figure 7 and Figure 8). In the paper, our narrative is that INVs should have cargo, fusion machinery and Rabs in order to be considered functional. We set

these up as three criteria for functionality that we test and we now show evidence for each with endogenous protein.

Of course, the other big question is the actual function of TPD54. Addressing this is probably unrealistic in an initial manuscript, but this requires that the new phenomena in the paper are as robust and unambiguous as possible.

Reviewer #1 (Comments to the Authors (Required)):

We thank the Reviewer for their insightful comments on our previous manuscript. The revision experiments caused us to completely rethink our conclusion that TPD54 is an effector at multiple Rabs. While we still think that TPD54 binds to some Rabs we simply don't have evidence for the level of promiscuity we were claiming. The discovery of the new vesicle class (now referred to as INVs, intracellular nanovesicles) has been strengthened during our revisions and this is now the central focus of the manuscript.

This is an interesting contribution providing evidence that Tumor Protein D54 has effector-typical properties for a number of Rab proteins. Tumor Protein D54 is a member of the Tumor Protein D52 (TPD52) family, most of which are expressed ubiquitously, and at least in the case of TPD52 are highly expressed in several cancers. Their role remains fairly obscure, although they have been implicated in membrane trafficking. TPD54, the subject of this article, is particularly highly expressed.

Localisation studies of TPD54 reported here show association with the Golgi apparatus and endosomes, and diffuse distribution throughout the cytoplasm. Using Golgi and endosomal markers, the authors conclude that there is colocalization with TPD54, although inspection of Fig. 1 A suggests that this is better described as a localization to the same organelle rather than a colocalization, particularly for the Golgi marker (GalT). The authors should comment on this. Subsequent pulldown experiments provided evidence that TPD54 interacts with members of the TPD52 family (as expected, since they are known to dimerize and heterodimerize), but also with Rab2a, Rab5c and Rab14. This suggested a possible involvement with anterograde and recycling pathways, which was confirmed in a general using TPD54-depleted cells. It was already known that TPD52 interacts with Rab5c (quoted in the manuscript), and there was earlier indirect evidence of interaction of TPD52 with Rab5, Rab6 and Rab9 (Zhang, Ding and Barbieri, JBC (20016)) which should be cited in the paper.

We agree with the reviewer that TPD54 is in the same region as GalT, APPL1 etc. but not co-localized with them. In the manuscript we now say (underlined words are changed from previous version):

GFP-TPD54 fluorescence was apparently diffuse in the cytoplasm, but was also seen at the Golgi apparatus, marked with GalT-mCherry, and on endosomes, marked by APPL1 and OCRL1. It also partially overlaps with various membrane trafficking proteins, such as clathrin light chain a and the R-SNARE, VAMP2 (Figure 1A).

We have changed the legend for Figure 1 so that it no longer mentions colocalization.

We have included a citation to Zhang et al 2007 as well as the TPD52-Rab5c reference in the discussion section where we discuss possible interactions with Rabs.

Mislocation of TPD54 to mitochondria resulted in capture of small vesicles, affected mitochondria morphology and caused aggregation. Essentially all labelled TPD54 was rerouted to mitochondria, with the "cytosolic" fraction being recruited more rapidly than that on endosomes. Using a library of 43 labelled Rabs, the authors were then able to show that 16 Rabs relocated to mitochondria with TPD54, suggesting an interaction (direct or indirect). In a reciprocal relocation experiment in which 3 Rabs (Rab11a, Rab 25 and Rab7a) were recruited to mitochondria, TPD54 was also mislocated in the case of Rab11a and Rab25, which were both positive in the "forward" screen, but not in the case of Rab7, which was negative in the first screen, confirming the general interpretation of interaction between the 16 Rabs (or at least the vesicles on which they are located) and TPD54. Analysis of the group of Rabs identified suggested that they are all involved in anterograde or recycling

pathways, although not all Rabs known to be involved on these pathways were identified as TPD54 interactors.

Direct evidence for interaction of Rab1a in its active form (using GppNHp) with TPD54 was obtained using recombinant proteins, whereas the inactive (GDP) form did not interact. Rab1a was positive in the rerouting experiment. Rab6a, which was negative in the screen did not interact. Thus, at least in the case of Rab1a, TPD54 has the properties of an effector. Ideally, it would have been good to have a few more positive and negative examples to give the conclusions a solid basis. One more result helps in this respect: using "active" and "inactive" mutants of Rab 30, Rab1a and Rab6a, only the active (or wild type) variants of Rab30 and Rab1a could be rerouted to mitochondria, while the inactive mutants and none of the Rab6 variants relocated.

In addressing this point, which was also raised by all Reviewers, we realized that our conclusion that TPD54 was a promiscuous Rab effector was premature. We have changed the focus of the paper as a result. A brief summary of the work we did:

- We cloned a further 13 GST-tagged Rabs for this revision work (Rab7a, Rab9a, Rab10, Rab11a, Rab12, Rab14, Rab19, Rab21, Rab25, Rab26, Rab30, Rab33b and Rab43).
- TPD54 binding experiments were done with Rab7a, Rab11a, Rab12, Rab21, Rab26, Rab30 and Rab33b. Expression of the remaining Rabs was insufficient for binding assays (Rab9a, Rab10, Rab14, Rab19, Rab25 and Rab43).
- Again, we tested for "effector binding". Does TPD54 bind preferentially to the active (GppNHp-bound) form of the Rab compared to the inactive (GDP-bound)?
- The results were more variable than what we observed with Rab6a and Rab1a, shown in the previous manuscript.
- TPD54 bound to Rab11a, Rab12 and Rab33b, but showed no preference between the active and inactive forms (n=2-3). These Rabs were positive in our screen.
- More TPD54 bound to the inactive forms of Rab26 and Rab30 than the active forms (n=2). These Rabs were positive in our screen.
- TPD54 was seen to bind to Rab7a and Rab21 (n=1). These Rabs were negative in our screen. Here there was more binding of the inactive forms of Rab7a and Rab21 compared to the active forms.

Our conclusion is that we got "lucky" with our experiments on Rab1a/Rab6a and this misled us into thinking that positive hits in our Rab screen were due to TPD54 being an effector at all of the co-rerouted Rabs. Now that we have tested a wider variety of Rabs in the same way, we can see that this extrapolation is not valid.

In the revised paper we have removed the TPD54/Rab binding experiments and we have changed the focus to be on the vesicles themselves rather than on TPD54. In the discussion we leave open the possibility that TPD54 binds Rabs. After all, we are still showing the mass spec data that TPD54 binds Rab2a, Rab5c, Rab14. However, the conclusion about the Rab screen is parsimonious: positive Rabs are passengers on the rerouted INVs with no conclusion possible about direct binding.

Finally, evidence is given that TPD54 is active in promoting vesicle fusion, since knock-down or knock-out of TPD54 led to dispersion of the trans-Golgi network. This interpretation would need considerable further evidence to allow a definite conclusion with mechanistic insights on this question.

The speculation on a role in membrane fusion has been removed.

In general, the main conclusion of the paper (that TPD54 is a promiscuous Rab effector) appears to be well founded, although more biochemical evidence would make the argument more compelling. One point that perhaps needs some clarifying arguments is that made in

the second sentence of the discussion. Did the authors really show that TPD54 is present on numerous (types of) vesicles? Is this not an interpretation of the evidence that at least 16 Rabs are relocated with TPD54 to mitochondria? Do we know how much of this is due to vesicle recruitment, and how much might come from recruitment of a genuine cytosolic Rab fraction (presumably GDI bound)? Perhaps there are arguments that I have missed on these points. This is related to the commonly arising question of "what recruits what" in terms of Rab composition of membranes and the general principle of membrane identity.

This is an excellent point. In short, we think that any co-rerouting is due to Rabs, SNAREs or cargo behaving as passengers. Or at least we cannot conclude that a co-rerouted protein binds to TPD54. We were being vague on this point in the previous version, as the Reviewer has identified. Our experiments suggest that the cytosolic (non-membrane bound) pool of TPD54 is minimal. So, when we do rerouting, almost all the signal comes from relocation of INVs and any passenger proteins, and not from cytosolic TPD54 and anything bound to it. This is because the kinetics of rerouting follow a single process on the order of tens of seconds, similar to our FRAP data which also only shows a minimal cytosolic pool. Since we now think it is unlikely that TPD54 binds all of the Rabs in our screen, the most parsimonious explanation for our results is that TPD54 localizes independently to INVs and lots of different Rabs can travel on them. Accordingly:

- We now state this interpretation in the manuscript
- We have removed the schematic rerouting diagrams which were ambiguous
- At the start of the discussion, we still say "TPD54 [...] is found on numerous small vesicles", but it is clear we mean that there are lots of them since the manuscript is describing them as a single class of vesicle. We go on to describe exactly what we mean.

Reviewer #2 (Comments to the Authors (Required)):

We thank the Reviewer for their time and comments on the previous version of our manuscript. We were pleased that they thought that our work is of interest to the JCB readership. Their review highlighted to us that the small vesicles, which we now term INVs (intracellular nanovesicles), are the most exciting aspect of our paper. This chimes well with feedback we have received from the community on our preprint. This Reviewer's challenge to us to obtain more evidence for the existence of INVs was a major factor in changing the focus of our paper, as described below.

The authors study the abundant but mechanistically uncharacterized protein TPD54. Their principal conclusions are as follows. (1) TPD54 associates with vesicles. (2) These vesicles are not obviously coated and are very small, with a diameter of about 30 nm. (3) These vesicles contain some cargoes (e.g., with dileucine sorting motifs) but not others. (4) These vesicles contain some SNAREs (e.g., VAMPs 2, 3, 7, and 8) but not others (VAMP4, five tested Q-SNAREs). (5) TPD54 interacts with at least 16 Rabs. (6) In the case of Rab1a, this interaction is direct. (7) In the cases of Rab30 and Rab1a, this interaction is specific to the active, GTP-bound form of the Rab. (8) Depleting TPD54 leads to anterograde and recycling trafficking defects and Golgi dispersal.

There are a number of surprises here. First, the vesicles. They are smaller than well-characterized vesicular carriers (e.g., COPI, COPII, clathrin-coated vesicles) and the authors boldly propose that they are an overlooked class of vesicles that have heretofore evaded study. Second, the large number of VAMPs that appear to co-populate TPD54 vesicles. This, along with the functional consequences of depleting TPD54, suggests that TPD54 vesicles are involved in trafficking to various destinations, making it all the more remarkable that they have not previously been reported. Third, the promiscuous Rab interactome. Given the modest size of TPD54, it's rather incredible that it is a potential effector for over a dozen Rabs. This suggests a key conserved function. What is it, and how has it escaped attention until now? All of which is to say that there is a much of potential interest to the reader of JCB in this manuscript.

Major points

1. The main evidence for Rab-TPD54 interactions comes from the observation that rerouting TPD54 to the mitochondria causes significant rerouting of 16 different Rabs. The cartoon in Fig. 3D suggests that the interaction between TPD54 and the rerouted Rab is direct, but it could also be indirect - any protein associated with the rerouted vesicle would presumably be rerouted along with it. The only evidence in favor of a direct interaction is the pull-down assay (Fig. 9a), and it only contains one positive result (for GppNHp-loaded GST-Rab1a). Given the major focus of this paper on TPD54 as a Rab effector, and the mechanistic consequences thereof, it would be very reassuring to have further evidence for direct interactions.

In addressing this point, which was also raised by all Reviewers, we realized that our conclusion that TPD54 was a promiscuous Rab effector was premature. We have changed the focus of the paper as a result. A brief summary of the work we did:

- We cloned a further 13 GST-tagged Rabs for this revision work (Rab7a, Rab9a, Rab10, Rab11a, Rab12, Rab14, Rab19, Rab21, Rab25, Rab26, Rab30, Rab33b and Rab43).
- TPD54 binding experiments were done with Rab7a, Rab11a, Rab12, Rab21, Rab26, Rab30 and Rab33b. Expression of the remaining Rabs was insufficient for binding assays (Rab9a, Rab10, Rab14, Rab19, Rab25 and Rab43).

- Again, we tested for “effector binding”. Does TPD54 bind preferentially to the active (GppNHp-bound) form of the Rab compared to the inactive (GDP-bound)?
- The results were more variable than what we observed with Rab6a and Rab1a, shown in the previous manuscript.
- TPD54 bound to Rab11a, Rab12 and Rab33b, but showed no preference between the active and inactive forms (n=2-3). These Rabs were positive in our screen.
- More TPD54 bound to the inactive forms of Rab26 and Rab30 than the active forms (n=2). These Rabs were positive in our screen.
- TPD54 was seen to bind to Rab7a and Rab21 (n=1). These Rabs were negative in our screen. Here there was more binding of the inactive forms of Rab7a and Rab21 compared to the active forms.

Our conclusion is that we got “lucky” with our experiments on Rab1a/Rab6a and this misled us into thinking that positive hits in our Rab screen were due to TPD54 being an effector at all of the co-rerouted Rabs. Now that we have tested a wider variety of Rabs in the same way, we can see that this extrapolation is not valid.

In the revised paper we have removed the TPD54/Rab binding experiments and we have changed the focus to be on the vesicles themselves rather than on TPD54. In the discussion we leave open the possibility that TPD54 binds Rabs. After all, we are still showing the mass spec data that TPD54 binds Rab2a, Rab5c, Rab14. However, the conclusion about the Rab screen is parsimonious: positive Rabs are passengers on the rerouted INVs with no conclusion possible about direct binding.

2. The striking (and central to this manuscript) idea that swarms of previously undetected 30-nm vesicles are conveying cargo hither and yon within the cell would greatly benefit from further direct experimental support. An 'obvious' idea would be immunogold labeling, in conjunction with rerouting experiments.

We have done super-resolution imaging in normal, unperturbed cells. By STORM we see TPD54-positive puncta with a size distribution that agrees with that observed in our vesicle capture-EM work. The average spot by STORM has a FWHM of 33.6 nm. This data is now in Figure 6 of the manuscript. In addition, we also measure the ‘flicker’ noted previously, using a simple image analysis method, to show the “swarms of vesicles” are indeed present in normal cells. We pursued these super-res and live cell imaging rather than immunogold since they are orthogonal to our original EM evidence.

We would like to thank the reviewer for this comment. The work we have done to meet this challenge has changed the direction of the manuscript and given us further confidence in our claim to have found a new class of intracellular vesicle.

Minor points

1. As GFP-tagging can significantly influence the intracellular localization of proteins, it would be reassuring to see that smaller tags (e.g. HA, flag) lead to the same localization.

We have now added a figure (Supplementary Figure S2) which shows FLAG-tagged TPD54 by immunofluorescence. Its subcellular distribution is similar to GFP-TPD54 or mCherry-TPD54. This is now mentioned in the first Results section.

2. The transferrin uptake/recycling assay should be presented more clearly; for example, which parts of Fig. 2h correspond to what phase of the assay?

We have added labels to indicate which parts correspond to uptake and recycling.

3. As the authors suggest that (most of the) TPD54 is actually not cytoplasmic but instead bound to small vesicles, does the initial rerouting of cytoplasmic mCherry-FKBP-TPD54 (and the deduced kinetics thereof) represent an overexpression artifact?

We don't think so. From the results we have, it does appear that most of the TPD54 is bound to small vesicles and that the cytosolic TPD54 pool is very small, although this does get bigger with overexpression (presumably the total surface of INVs is saturable). The kinetics of rerouting were best described as a single process on the order of tens of seconds. Since we know this corresponds to the rerouting of vesicles, and we don't detect any faster component, then any rerouting of a cytosolic pool must be pretty minimal. The manuscript is now unequivocal on this point.

4. I didn't entirely understand the 'flickering' argument for TPD54 being on vesicles, and in any case it seems that the cited video (SV4) actually represents the FRAP experiment discussed subsequently.

Apologies that this was unclear. The 'flickering' behavior is definitely descriptive. What we mean is that there are lots of individual vesicles moving around rapidly, causing a flickering effect. Because this is a descriptive term, we have now used an image analysis method based on spatiotemporal variance of fast capture live-cell imaging movies to measure this behavior quantitatively. The results of this are shown together with the STORM imaging in Figure 6 (parts A and B).

Previously, the movie SV4 was a "two-for-one" showing FRAP *and* flickering in the same movie. We now have a dedicated movie SV4 to show flickering fluorescence that has been analyzed in the Figure 6. FRAP is a separate movie (SV5).

Reviewer #3 (Comments to the Authors (Required)):

We thank the Reviewer for their time and their comments on our manuscript. Obviously, we are disappointed that the Reviewer got the impression that our manuscript was put together “carelessly” or “sloppily”. All authors have worked hard on this project for a long time and this criticism is difficult to digest for the authors on the paper who are early in their career.

Larocque *et al* investigate the role of tumour protein D54, member of the TPD52-like protein family, concerning its function in anterograde membrane trafficking and endosomal recycling. Using a wide range of cell biological tools, such as endogenous protein tagging, CRISPR-mediated KO cell lines, co-immunoprecipitation and rapamycin inducible mis-localization, they show that 1) several distinct Rab GTPases co-immunoprecipitated with TPD54 and 2) mis-localization of TPD54 to mitochondria lead to an accumulation of small vesicles on mitochondria concomitant with rerouting of different Rab GTPases to mitochondria. Combining the cell biological approach with biochemistry and EM the authors hypothesize that TPD54 associates with an unknown class of small vesicles and directly interacts with different Rab GTPases, constituting a promiscuous Rab effector.

While the key message of the study is provocative, the manuscript gives the impression of pieces of data put together in a rather careless and sloppy fashion. There are flaws and weaknesses at different levels, starting from the overall composition, which lacks coherence. On several occasions the work is inconsistent and it is not well written. Besides the presentation, I have serious concerns about the scientific quality of the data. The biochemistry on which the whole story is built is weak and incomplete. Furthermore, there is lack of reasoning, explanation and experimental care in different parts of the manuscript. For example, important controls are missing. The interpretation of the results seems to be biased towards the central hypothesis on several occasions, neglecting the possibility that serious perturbations such as overexpression of proteins and RNAi-mediated knock-down may also lead to the observation of off-target effects and overexpression artefacts. The extent of work and the number experiments needed for this manuscript to form a coherent, comprehensive and scientifically solid paper is far beyond a normal revision, I therefore recommend rejection of this manuscript.

Major comments

1. The title of the manuscript "Tumor Protein D54 is a promiscuous Rab effector" comprises a strong statement about the direct and nucleotide dependent interaction of TPD54 with Rab GTPases which the authors do not convincingly show anywhere in the manuscript. Biochemical indication is only provided for the interaction of TPD54 with Rab1a using a GST pulldown assay. For other Rabs the authors only show co-rerouting experiments, which are not sufficient to prove a direct and nucleotide dependent interaction and hence are not suitable to identify Rab effectors. Even though the requirements for Rab effectors are clearly mentioned in the manuscript, the authors base their statement on a single interaction assay with Rab1a and co-rerouting experiments. To prove this hypothesis, interaction assays for most if not the full set of Rab effectors tested in the co-rerouting experiments combined with a thorough biochemical characterization of the components (see point 2) would be necessary.

The Reviewer is absolutely correct on this point. Our central hypothesis needed more rigorous biochemical interrogation.

In addressing this point, which was also raised by all Reviewers, we realized that our conclusion that TPD54 was a promiscuous Rab effector was premature. We have changed the focus of the paper as a result. A brief summary of the work we did:

- We cloned a further 13 GST-tagged Rabs for this revision work (Rab7a, Rab9a, Rab10, Rab11a, Rab12, Rab14, Rab19, Rab21, Rab25, Rab26, Rab30, Rab33b and Rab43).
- TPD54 binding experiments were done with Rab7a, Rab11a, Rab12, Rab21, Rab26, Rab30 and Rab33b. Expression of the remaining Rabs was insufficient for binding assays (Rab9a, Rab10, Rab14, Rab19, Rab25 and Rab43).
- Again, we tested for “effector binding”. Does TPD54 bind preferentially to the active (GppNHp-bound) form of the Rab compared to the inactive (GDP-bound)?
- The results were more variable than what we observed with Rab6a and Rab1a, shown in the previous manuscript.
- TPD54 bound to Rab11a, Rab12 and Rab33b, but showed no preference between the active and inactive forms (n=2-3). These Rabs were positive in our screen.
- More TPD54 bound to the inactive forms of Rab26 and Rab30 than the active forms (n=2). These Rabs were positive in our screen.
- TPD54 was seen to bind to Rab7a and Rab21 (n=1). These Rabs were negative in our screen. Here there was more binding of the inactive forms of Rab7a and Rab21 compared to the active forms.

Our conclusion is that we got “lucky” with our experiments on Rab1a/Rab6a and this misled us into thinking that positive hits in our Rab screen were due to TPD54 being an effector at all of the co-rerouted Rabs. Now that we have tested a wider variety of Rabs in the same way, we can see that this extrapolation is not valid.

In the revised paper we have removed the TPD54/Rab binding experiments and we have changed the focus to be on the vesicles themselves rather than on TPD54. In the discussion we leave open the possibility that TPD54 binds Rabs. After all, we are still showing the mass spec data that TPD54 binds Rab2a, Rab5c, Rab14. However, the conclusion about the Rab screen is parsimonious: positive Rabs are passengers on the rerouted INVs with no conclusion possible about direct binding.

2. The interaction assays provided in the manuscript seem to be weak, erroneous in experimental details and not properly described in the methods section.

Firstly, why do the authors use an MBP-TPD54-His variant of the protein to perform the interaction assay, even though the MBP tag is not necessary for the experiment? Details about the construct design are omitted in the manuscript and should be clarified. As the MBP tag is often used to keep proteins soluble, a thorough biochemical characterization is required to ensure protein quality.

Secondly, the authors state that “proteins were eluted from the beads in Laemmli buffer...”. This experimental approach potentially leads to a misinterpretation of the results, as Laemmli buffer also elutes proteins that are aggregated on the beads, instead of interacting with the GST-Rab. This can be avoided by eluting with a Glutathione containing elution buffer (20 mM GSH) and this needs to be shown to strengthen the authors' point.

Next problem, after purification of GST, GST-Rab1a, GST-Rab6a and MBP-TPD54-His a buffer exchange is required to perform the described experiments (GST interaction assays). However, this step is not described in the methods section nor mentioned in the manuscript. Was it done by dialysis, size exclusion chromatography or a different method?

How was the successful loading of the Rab GTPases checked? The authors do not provide any detail of how quantitative loading was ensured.

All these points are especially important as the main claim of the paper is the direct interaction of TPD54 with different Rab GTPases as an effector, i.e. in a nucleotide-dependent manner.

These criticisms are all excellent, valid and greatly informed our revision work. However as described above, we have now removed all of the binding data and changed the emphasis of the manuscript.

3. Was the GFP-TPD54 cell line described in the first section of the results tested using the transferrin uptake assay mentioned in the second paragraph of the results? If not please check if the GFP-TPD54 cell behaves similarly as HeLa wt.

We use the knock-in cell line to assess the cellular localization of TPD54 rather than function. However, the question of whether the cell line is phenotypically similar to the parental line is valid because this may affect localization. To assess if the localization of the cell line is normal, we have added a figure (Supplementary Figure S2) that shows that the localization is similar to that seen following expression of TPD54 tagged with GFP, mCherry or FLAG.

4. How many different siRNAs against TPD54 were used in the knock-down experiments (Figure 2)? How many of these gave a consistent phenotype? The authors need to provide evidence that the phenotype is not due to off-target effects of the siRNA.

We are sure that the phenotypes we observe are not due to off-target effects of the siRNA. The same RUSH phenotype is seen with three siRNAs and these data are now shown in Figure 2. We showed that the Golgi integrity phenotype is present by siRNA and by KO, and that it can be rescued by re-expression of TPD54. From other unpublished work in the lab we know that depletion of TPD54 leads to defects in cell migration and that this is caused by changes in integrin recycling. This phenotype is the same with three different siRNAs and can be rescued by re-expression of TPD54. This data also corroborates the defect in recycling shown in Figure 2 for transferrin recycling.

5. The authors argue that a flickering in the cytoplasmic TPD54 signal is observed in contrast to GFP. After a quantification of diffusion rates using FRAP it is stated that "These experiments indicate that the freely diffusing pool of TPD54 is minimal and the majority of TPD54 is associated with small vesicles below the resolution limit." Would the authors please explain why TPD54 needs to be bound to small vesicles and what would be the result of the FRAP experiment if TPD54 were to unspecifically interact with any membrane?

The result would be identical. Strictly speaking, we should have said "majority of TPD54 is associated with structures below the resolution limit". However, given that we had evidence that TPD54 is on small vesicles, we made the connection. Regardless, the manuscript is now stronger on this point due to super-resolution imaging (see below).

Can these small vesicles be visualized by EM under normal conditions? The overexpression of mCherry-FKBP-TPD54 may induce the formation of these vesicles which are subsequently targeted to mitochondria by rapamycin.

This point was also raised by Reviewer #2. We have done super-resolution imaging in normal, unperturbed cells. By STORM imaging we see TPD54-positive puncta with a size distribution that agrees with that observed in our vesicle capture-EM work. The average spot by STORM has a FWHM of 33.6 nm. These results, obtained with no expression nor rapamycin, are shown in Figure 6.

6. Following the observation of small vesicles on mitochondria, the authors assess their functionality using co-rerouting of overexpressed model cargo and co-localization of SNARE proteins. How do these experiments show that the vesicles are functional? What is the function of these vesicles? The authors indicate in the discussion that these vesicles are a so far un-described and unidentified. If true, it is necessary to identify these vesicles in other

cell lines under normal conditions and to further characterize them, otherwise this idea stands in the air without any significance.

In the previous version, we had captured small vesicles on mitochondria. These might have been fragments of membrane rather than functional vesicles. We now have other evidence that these vesicles are found in normal cells (by STORM and live cell imaging) so this is less of a concern. However, it is still possible that these little vesicles don't do anything, i.e. are non-functional. We reason that, for a vesicle to be considered functional, it must a) carry cargo (otherwise it is not a transport vesicle), b) have fusion machinery (otherwise it cannot deliver the cargo), and c) have a Rab (for transport or other function). We test these criteria and find that INVs meet all of these criteria and so they are functional by this definition (Figure 7AB, Figure 8, Figure 9).

Related to functionality, we also show that cargo is actively trafficked from the cell surface via INVs, further showing that these vesicles are used during usual trafficking of transmembrane cargo (Figure 7CD).

In addition, we show that reduction of TPD54 affects anterograde traffic, recycling and Golgi integrity which further highlights the function of INVs (Figure 2 and Figure 3). These data were all in the previous manuscript but are now bundled together in the same results section rather than being split.

We apologize that we had not made it clear to the Reviewer what we meant by functionality. It is now more explicitly stated what we mean in the manuscript. There is now a Results section entitled "Intracellular nanovesicles meet three criteria for functionality". Which starts by saying:

The small size of INVs raised the question of whether or not they were functional. We reasoned that there are three basic criteria for a vesicle to be considered functional. It must i) contain cargo, ii) have fusion machinery, and iii) associate with a Rab GTPase.

What is the situation in non-cancer cell lines where TPD54 is not overexpressed, as observed in HeLa cells? The experiments are not very well explained and it seems to me that the experimental design is not suitable to answer the questions posed in the manuscript.

The Reviewer is conflating two separate issues here. First, the striking abundance of TPD54 in HeLa cells reported by Matthias Mann's lab. Second, TPD52 is overexpressed in cancer (many papers on this) and there is some evidence that TPD53 and TPD54 are also overexpressed in human cancers (fewer papers). It is unknown whether the abundance in HeLa is because of cancerous overexpression.

In fact, the human protein atlas shows that the amount of TPD54 RNA in HeLa is similar to most other cell lines (see Reviewer Figure R2A). This means that the abundance of TPD54 reflects high cellular expression in normal cells rather than gene duplication or some other genomic rearrangement in HeLa cells. We checked by western blot the expression of TPD54 in hTert-RPE1 cells (stable diploid human retinal epithelial cells) and it is similar to that seen in HeLa (Reviewer Figure R2B).

7. How does overexpression of Rab GTPases affect the rerouting to mitochondria upon addition of rapamycin? Is the rerouting still observed in cell lines expressing near endogenous levels of fluorescent Rab GTPases?

This is an interesting question although the answer is less relevant to the new manuscript. We have added a new figure (Supplementary Figure S6) that shows the extent of rerouting of endogenous Rab1a. The kinetics of TPD54 rerouting in cells expressing only mCherry-FKBP-TPD54 or expressing mCherry-FKBP-TPD54 with a GFP-Rab is not significantly altered. We tested Rab1a, Rab5a, Rab6a and Rab30 and saw kinetics of 30-40 s (τ) in all cases.

8. "Second, we marked up our hits on a phylogenetic tree of the Rab collection (Supplementary Figure S4B). This diagram showed that the positive hits generally belonged to a clade, with a common ancestral sequence. This suggested to us that TPD54 binds directly to these particular Rab GTPases rather than being independently localized on the same vesicle." Can the authors please explain how a common ancestral sequence points towards a direct interaction between TPD54 and a set of Rab GTPases? Do the authors implicate the common ancestral sequence to be the interface? Has this been verified by mutational studies?

This has been removed from the manuscript.

9. In the last section of the results it is not clear how a dispersal phenotype of the Golgi indicates a role of TPD54 in vesicle fusion. The authors need to explain their line of argumentation and provide further evidence for their hypothesis.

In the revised version of the paper, all of the phenotypic characterization is bundled together in Figure 2 and Figure 3. The emphasis of the paper is now on the discovery of the new vesicle type. The depletion of TPD54, which is our only molecular handle on these vesicles, causes a range of defects in anterograde traffic, recycling, and on Golgi integrity. The purpose of this paper is therefore not to dive into why TPD54 causes any one of these phenotypes, it is to report the discovery of this vesicle type and to show that they participate in a range of pathways.

Minor:

1. The manuscript is written in a rather sloppy and partly unscientific fashion. The introduction reads very general as small chapters summarizing textbook knowledge, rather than a dedicated introduction to this specific paper. Some examples:

"Despite its abundance, there are virtually no published data on the cell biology of TPD54." What does that mean? If there is no data published it can be stated like that, if there is data please cite.

We mean that there aren't many papers on TPD54 and those that have been published and feature a bit of cell biology, aren't very good. There are only 30 papers on TPD54 on PubMed and 4 of those have been retracted! It's not true to say that there are none, but there are none worth citing.

"To our frustration, the Golgi dispersal phenotype in both knockout clones disappeared with repeated passaging, which might be explained by compensation for the chronic loss of TPD54 in knockout cells. These experiments confirm that the dispersal phenotype is due to loss of TPD54 specifically and is not the result of off-target action. Furthermore, they indicate that the effector function for TPD54 is likely to be in the promotion of vesicle fusion." What does it mean if the phenotype is lost? Can the other TPD52-like proteins compensate the effect?

Yes, compensation for a loss of expression is well known in many studies on knockout mice. There are recent studies on giantin CRISPR KO cell lines which have explored the changes in gene expression that occur in response to the loss of this Golgi protein (Stevenson *et al.*, 2017 JCS, PMID 29093022). TPD54 has two very close homologs in TPD52 and TPD53. It is likely that upregulation of these proteins compensates for loss of TPD54.

We don't think what we wrote was "non-scientific" if this is what the Reviewer is referring to. In fact, we endeavor to write in an engaging style in contrast to the majority of papers that we find too dry or rather boring to read. Hopefully the reviewer can forgive us for expressing our frustration at the disappearance of this nice phenotype in the KO cell line.

2. A figure with a schematic alignment of the different TPD52-like proteins would be helpful

Since we are not doing any molecular dissection of TPD54 or a comparison with other TPD52-like proteins, we don't think it would be helpful to add such an alignment.

3. Why do the authors investigate the interaction of TPD54 with Rab1 by GST pulldown assays, when Rab14, Rab2a and Rab5c were identified as significant hits by co-immunoprecipitation?

This data is now removed from the paper.

September 30, 2019

RE: JCB Manuscript #201812044R-A

Prof. Stephen J Royle
University of Warwick
Centre for Mechanochemical Cell Biology Division of Biomedical Cell Biology Warwick Medical School
Gibbet Hill Road
Coventry CV4 7AL
United Kingdom

Dear Prof. Royle,

Thank you for submitting your revised manuscript entitled "Tumor Protein D54 defines a new class of intracellular transport vesicle". Both original Reviewers #1 and #2 and new Referee #4 (as discussed, brought in to help us evaluate the suitability of the work for JCB given the major change in its scope and time elapsed, consistent with JCB policy) find the revision very interesting and are supportive of publication. We would be happy to publish your paper in JCB pending final revisions necessary to meet our formatting guidelines (see details below) and pending edits to address Reviewer #3's suggestions.

- 1) Text limits: Character count for Articles and Tools is < 40,000, not including spaces. Count includes title page, abstract, introduction, results, discussion, acknowledgments, and figure legends. Count does not include materials and methods, references, tables, or supplemental legends.
- 2) JCB Articles can have up to 10 main and 5 supplemental figures. Each figure can span up to one entire page but all panels must fit on one page. Please rearrange the supplementary data to meet this limit.
- 3) Titles, eTOC: Please consider the following revision suggestions aimed at increasing the accessibility of the work for a broad audience and non-experts.

- Title: Tumor Protein D54 defines a new class of intracellular transport vesicles (plural)

- Running title (50 characters max, including spaces): We suggest making the running title more specific to the advance in the work:
"New small transport vesicles defined by TPD54"

- eTOC summary: A 40-word summary that describes the context and significance of the findings for a general readership should be included on the title page. The statement should be written in the present tense and refer to the work in the third person.
The summary statement "Larocque et al. discover a new class of transport vesicles in human cells termed intracellular nanovesicles. These small carriers are defined by the presence of TPD54, an

abundant yet poorly characterized protein." looks great, **please be sure to include it on the title page of the revised manuscript along with the running title.**

4) Figure formatting: Scale bars must be present on all microscopy images, including inset magnifications. Please add scale bars to 1A (magnifications), 4C (mags), 5B (mags), 7D (mags), 8 (mags), S2 (mags), S5 (mags), S6 (mags), S7 (mags), images in SV4 (mags)

5) Statistical analysis: Error bars on graphic representations of numerical data must be clearly described in the figure legend. The number of independent data points (n) represented in a graph must be indicated in the legend. Statistical methods should be explained in full in the materials and methods. For figures presenting pooled data the statistical measure should be defined in the figure legends.

Please indicate n/sample size/how many experiments the data are representative of: 2EFGH, 3BD, 5DE, 6BF, S4

6) Materials and methods: Should be comprehensive and not simply reference a previous publication for details on how an experiment was performed. Please provide full descriptions in the text for readers who may not have access to referenced manuscripts.

- Please describe all constructs, vectors, chimeras -- even if gifted to you by other investigators and described in other published papers -- or provide Addgene/catalog numbers.

- Please include *all* siRNA and CRISPR oligo sequences -- even controls, if available to you from the manufacturers.

- Microscope image acquisition: The following information must be provided about the acquisition and processing of images:

a. Make and model of microscope

b. Type, magnification, and numerical aperture of the objective lenses

c. Temperature

d. imaging medium

e. Fluorochromes

f. Camera make and model

g. Acquisition software

h. Any software used for image processing subsequent to data acquisition. Please include details and types of operations involved (e.g., type of deconvolution, 3D reconstitutions, surface or volume rendering, gamma adjustments, etc.).

7) References: There is no limit to the number of references cited in a manuscript. References should be cited parenthetically in the text by author and year of publication. Abbreviate the names of journals according to PubMed.

- Please note our formatting guidelines for preprint citations and correct both the ref list and in-text citations:

<http://jcb.rupress.org/reference-guidelines>

8) A summary paragraph of all supplemental material should appear at the end of the Materials and methods section.

- Please include one brief descriptive sentence per supplemental item.

A. MANUSCRIPT ORGANIZATION AND FORMATTING:

Full guidelines are available on our Instructions for Authors page, <http://jcb.rupress.org/submission-guidelines#revised>. **Submission of a paper that does not conform to JCB guidelines will delay the

acceptance of your manuscript.**

B. FINAL FILES:

-- High-resolution figure and video files: See our detailed guidelines for preparing your production-ready images, <http://jcb.rupress.org/fig-vid-guidelines>.

Thank you for this interesting contribution, we look forward to publishing your paper in the Journal of Cell Biology.

Sincerely,

Sean Munro, PhD
Monitoring Editor, Journal of Cell Biology

Melina Casadio, PhD
Senior Scientific Editor, Journal of Cell Biology

Reviewer #1 (Comments to the Authors (Required)):

The authors have taken the comments of the referees seriously and performed extensive additional experiments that add to the basis for the conclusions drawn. However, these conclusions are rather different from those in the original submission, as already seen in the title. These changes have been made on the basis of the referees' comments, and include a much more thorough and critical analysis of the results obtained previously and now additionally. I'm not sure I

like the emphasis on a "new" class of vesicles. They were presumably always there, so that they might be described as "newly discovered". Whether they really belong to a previously unrecognised class, or constitute a class not previously identified, or not, is not easy to judge, but on balance the hypothesis of a specific class of vesicles is probably justified.

Reviewer #2 (Comments to the Authors (Required)):

Clearly the reviewers took the comments of the reviewers very much to heart, and I believe that the resulting manuscript is substantially improved. In my opinion it is now suitable for publication in the JCB.

Reviewer #4 (Comments to the Authors (Required)):

This is a most intriguing manuscript. It presents a new class of intracellular vesicles marked by the enigmatic protein Tumor Protein D54. These vesicles are remarkable by their small size, by their role in various trafficking pathways at the TGN and endosome level and by the collection of Snares and Rab that they carry. The current paper is a re-submission of a previous manuscript, which was rejected. To not be biased by the previous round of reviews, I directly read the new manuscript and found it most interesting. Thereafter, I read the previous reviews. Clearly, the authors have written a new manuscript where the emphasis is put on the most intriguing part of the work: the identification of new transport vesicles. Overall, the experimental strategies use by the authors concur to the existence of Tumor Protein D54 vesicles; including FRAP data, super-resolution analysis and most notably a mitochondria-rerouting strategy coupled to Cargo + Snare analysis and electron microscopy. I recommend acceptance of this manuscript because even if I'm wondering why those vesicles have not been seen by others, I don't have any additional experiment to challenge this finding. The paper should raise a lot of interest (from skepticism to enthusiasm) by the cell biology community. The only point that I'm asking for is a zoom in of the vesicles as seen as a cement between mitochondria to actually see a membrane bound structure. At the currently scale, this is difficult to assess and it is known that 30 nm in diameter is close to the limit of a physically stable lipid bilayer structure.